# Promoting Exploration in Memory-Augmented Adam using Critical Momenta

**Pranshu Malviya**                                          *pranshu.malviya@mila.quebec*
*Mila - Quebec AI Institute, Polytechnique Montreal*

**Gonçalo Mordido**                                          *goncalomordido@gmail.com*
*Mila - Quebec AI Institute, Polytechnique Montreal*

**Aristide Baratin**                                          *baratina@mila.quebec*
*Samsung SAIT AI Lab Montreal*

**Reza Babanezhad Harikandeh**                               *babanezhad@gmail.com*
*Samsung SAIT AI Lab Montreal*

**Jerry Huang**                                              *jerry.huanng@mila.quebec*
*Mila - Quebec AI Institute, Université de Montréal*

**Simon Lacoste-Julien**                                     *slacoste@iro.umontreal.ca*
*Mila - Quebec AI Institute, Université de Montréal, Samsung SAIT AI Lab Montreal, Canada CIFAR AI Chair*

**Razvan Pascanu**                                           *razp@google.com*
*Google DeepMind*

**Sarath Chandar**                                           *sarath.chandar@mila.quebec*
*Mila - Quebec AI Institute, Polytechnique Montreal, Canada CIFAR AI Chair*

**Reviewed on OpenReview:** *https://openreview.net/forum?id=sHSkJqyQgW*

## Abstract

Adaptive gradient-based optimizers, notably Adam, have left their mark in training large-scale deep learning models, offering fast convergence and robustness to hyperparameter settings. However, they often struggle with generalization, attributed to their tendency to converge to sharp minima in the loss landscape. To address this, we propose a new memory-augmented version of Adam that encourages exploration towards flatter minima by incorporating a buffer of critical momentum terms during training. This buffer prompts the optimizer to overshoot beyond narrow minima, promoting exploration. Through comprehensive analysis in simple settings, we illustrate the efficacy of our approach in increasing exploration and bias towards flatter minima. We empirically demonstrate that it can improve model performance for image classification on ImageNet and CIFAR10/100, language modelling on Penn Treebank, and online learning tasks on TinyImageNet and 5-dataset. Our code is available at `https://github.com/chandar-lab/CMOptimizer`.

## 1 Introduction

Deep learning models are often sensitive to the choice of optimizer used during training, which significantly influences convergence speed and the qualitative properties of the minima to which the system converges (Choi et al., 2019). Stochastic gradient descent (SGD) (Robbins & Monro, 1951), SGD with momentum (Polyak, 1964), and adaptive gradient methods such as Adam (Kingma & Ba, 2015) have been the most popular choices for training large-scale models.

Adaptive gradient methods are advantageous as they automatically adjust the learning rate on a per-coordinate basis, converging quickly with minimal hyperparameter tuning by using information about the loss curvature. However, they are also known to achieve worse generalization performance than SGD (Wilson et al., 2017; Zhou et al., 2020; Zou et al., 2023), which several recent works suggest is due to the greater stability of adaptive optimizers (Zhou et al., 2020; Wu et al., 2018a; Cohen et al., 2022). This can lead the system to converge to sharper minima than SGD, resulting in worse generalization performance (Hochreiter & Schmidhuber, 1994; Keskar et al., 2016; Dziugaite & Roy, 2017; Neyshabur et al., 2017; Chaudhari et al., 2017; Izmailov et al., 2018; Kaur et al., 2023).

However, similar to exploration in reinforcement learning, we hypothesize that equipping Adam with *an exploration strategy* could improve performance by escaping sharp minima. Building upon the framework proposed by McRae et al. (2022), which maintains a buffer containing a limited history of gradients from previous iterations (called *critical gradients* or CG) during training, the goal is to allow the optimizer to overshoot and escape sharp minima by adding inertia to the learning process, as to control for the necessary width of the minima in order for the system to converge. However, we show that the original memory-augmented adaptive optimizers proposed by McRae et al. (2022), particularly Adam using CG (referred to as Adam+CG), suffer from *gradient cancellation*: a phenomenon where new gradients have high directional variance and large norm around a sharp minima. This leads to the aggregated gradient over the buffer to vanish, preventing the optimizer from escaping sharp minima, which is in agreement with the poor generalization performance presented by McRae et al. (2022).

We propose to instead store critical momenta (CM) during training, leading to a new memory-augmented version of Adam (Algorithm 1) that can effectively escape sharp basins and converge to flat loss regions. Figure 1 illustrates the optimization trajectories, on a toy 2D loss surface corresponding to the Goldstein–Price (GP) function (Picheny et al., 2013), of Adam, Adam+CG, Adam+CM, and Adam combined with sharpness-aware minimization (Adam+SAM) (Foret et al., 2021) from different initialisations. We observe that while other optimizers converge to higher and often sharper loss regions, Adam+CM is able to find the flat region that contains the global minimum.

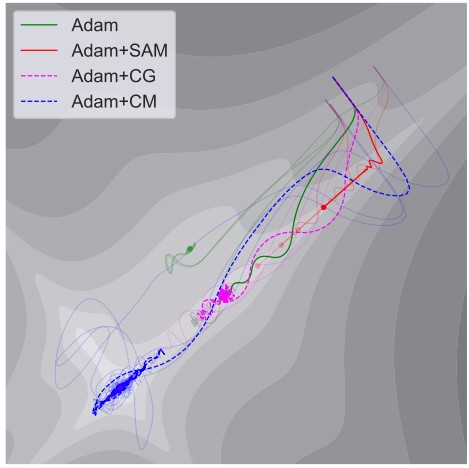

Algorithm 1: Adam with Critical Momenta

**Require:** Initial parameters $\theta_0$ and moments $m_0, v_0^M$, loss $L$, step size $\alpha$, buffer $\mathbf{m}_c$, capacity $C$, decay $\lambda$
    **for** $t = 1, 2, \cdots$ **do**
        Sample mini-batch & compute loss gradient
        Update 1st moments $m_t$ with equation 4
        Aggregate buffer moments $m_t^M \leftarrow m_t$ with equation 4
        Update 2nd moments $v_t^M$ with equation 5
        **if** buffer is not full **then**
            Add $m_t$ to $\mathbf{m}_c$
        **else if** Priority$(m_t) > \min($Priority$(\mathbf{m}_c))$ **then**
            Replace smallest priority element with $m_t$
        **end if**
        Decay Priority$(\mathbf{m}_c)$ using $\lambda$
        Update parameter $\theta_t$ with equation 7
    **end for**

Figure 1: (Left) Learning trajectories for different optimizers on the Goldstein-Price loss function starting from different initial points. While the other optimizers get stuck in sub-optimal surfaces, Adam+CM explores a lower loss surface and is able to reach the global minimum. (Right) Pseudo-code for Adam with critical momenta (Adam+CM).

The key contributions of our work are as follows:

- We introduce a framework for promoting exploration in adaptive optimizers (section 3). We propose a new memory-augmented version of Adam, which stores and leverages a buffer of critical momenta from previous iterations during training.

- We provide a theoretical convergence analysis of our method in simplified settings (subsection 3.2).

- Using numerous examples and benchmarks, we illustrate how our method surpasses existing memory-augmented methods and promotes exploration towards flat minima (section 4).

- We observe empirically an improvement in model performance in supervised and online learning settings (section 5).

## 2    Related work

To improve convergence speed and achieve better generalization in deep learning models, numerous optimizers have been proposed. While SGD with momentum tends to show superior performance in particular scenarios, it usually requires careful hyperparameter tuning (Le et al., 2011). On the other hand, adaptive optimization methods (Duchi et al., 2011; Hinton et al., 2012; Zeiler, 2012), which adjust the learning rate for each parameter based on past gradient information to accelerate convergence, have reached state-of-the-art performance in many supervised learning problems while being more robust to hyperparameter choice. In particular, Adam (Kingma & Ba, 2015) combines momentum with an adaptive learning rate and has become the preeminent choice of optimizer across a variety of models and tasks, particularly in large-scale deep learning models (Dozat, 2016; Vaswani et al., 2017). Several Adam variants have since been proposed (Loshchilov & Hutter, 2019; Zhuang et al., 2020; Granziol et al., 2020; Defazio & Jelassi, 2022) to tackle Adam's lack of generalization ability (Wu et al., 2018b; Zhou et al., 2020; Zou et al., 2023; Cohen et al., 2022).

Converging to flat minima has been shown to be a viable way of indirectly improving generalization performance (Hochreiter & Schmidhuber, 1994; Keskar et al., 2016; Dziugaite & Roy, 2017; Neyshabur et al., 2017; Izmailov et al., 2018; Kaur et al., 2023; Jiang et al., 2020). For example, sharpness-aware minimization (SAM) Foret et al. (2021) jointly maximizes model performance and minimizes sharpness within a specific neighborhood during training. Since its proposal, SAM has been utilized in several applications, enhancing generalization in vision transformers (Dosovitskiy et al., 2021; Chen et al., 2022), reducing quantization error (Liu et al., 2023), and improving model robustness (Mordido et al., 2022). Numerous methods have been proposed to further improve its generalization performance, *e.g.* by changing the neighborhood shape (Kim et al., 2022b) or reformulating the definition of sharpness (Kwon et al., 2021; Zhuang et al., 2022), and to reduce its cost, mostly focusing on alleviating the need for the double backward and forward passes required by the original algorithm (Du et al., 2022a;b; Liu et al., 2022).

Memory-augmented optimizers extend standard optimizers by storing gradient-based information during training to improve performance. Hence, they present a trade-off between performance and memory usage. Different memory augmentation optimization methods have distinct memory requirements. For instance, stochastic accelerated gradient (SAG) (Roux et al., 2012) and its adaptive variant, SAGA (Defazio et al., 2014), require storing all past gradients to achieve a faster convergence rate. While such methods show great performance benefits, their large memory requirements often make them impractical in the context of deep learning. On the other hand, one may only use a subset of past gradients, as proposed in limited-history BFGS (LBFGS) (Nocedal, 1980), its online variant (oLBFGS) (Schraudolph et al., 2007), and stochastic dual coordinate ascent (SDCA) (Shalev-Shwartz & Zhang, 2013). Additionally, memory-augmented frameworks with critical gradients (CG) use a fixed-sized gradient buffer during training, which has been shown to achieve a good performance and memory trade-off for deep learning compared to the previous methods (McRae et al., 2022).

In this work, we further improve upon CG by storing critical momenta instead of critical gradients, leading to a better exploration of the loss surface by adaptive optimizers, particularly Adam.

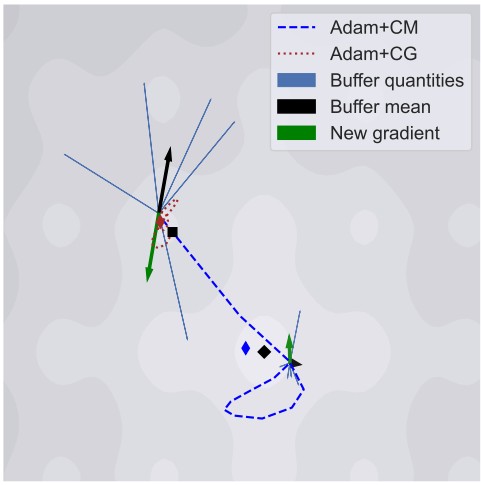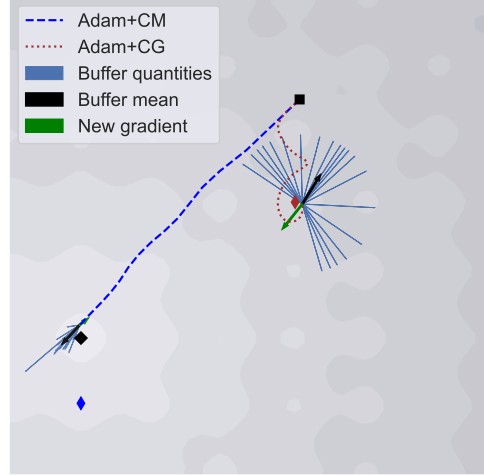

Figure 2: First 10 steps of Adam+CG and Adam+CM trajectories on Ackley loss surface. Coloured diamonds represent the final points reached by the optimizers. Gradient cancellation is observed in Adam+CG as buffer mean and new gradients cancel each other out, yielding a small update. Conversely, Adam+CM escapes sub-optimal minima and converges near the global minimum.

## 3 Memory-augmented Adam

We build upon the memory-augmented framework presented by McRae et al. (2022) and focus on Adam in a supervised learning setting. Adam (Kingma & Ba, 2015) has standard parameter updates

$$m_t = \beta_1 m_{t-1} + (1 - \beta_1)g_t, \quad v_t = \beta_2 v_{t-1} + (1 - \beta_2)g_t^2, \tag{1}$$

$$\hat{m}_t = \frac{m_t}{1 - \beta_1^t}, \quad \hat{v}_t = \frac{v_t}{1 - \beta_2^t}, \quad \theta_{t+1} = \theta_t - \alpha \frac{\hat{m}_t}{\sqrt{\hat{v}_t + \epsilon}}. \tag{2}$$

where $\theta_t$ denotes the model parameter at iteration $t$, $g_t$ is the loss gradient on the current mini-batch, $\alpha$ is the learning rate, $\beta_1, \beta_2 \in [0, 1)$ are the decay rates for $m_t$ and $v_t$.

**Critical gradients (CG).** To memory-augment Adam, McRae et al. (2022) introduce a fixed-size buffer $\mathbf{g}_c$ of priority gradients $g_c$ maintained in memory during training, and apply an aggregation function over this buffer to modify the moment updates (Equation 1):

$$m_t^G = \beta_1 m_{t-1}^G + (1 - \beta_1)\texttt{aggr}(g_t, \mathbf{g}_c), \quad v_t^G = \beta_2 v_{t-1}^G + (1 - \beta_2)\texttt{aggr}(g_t, \mathbf{g}_c)^2. \tag{3}$$

The gradient $l_2$-norm is used as the selection criterion for the buffer. The buffer takes the form of a dictionary where the key-value pairs are $(\|g_c\|_2, g_c)$; additionally, the priority keys are decayed at each iteration by a decay factor $\lambda \in (0, 1)$ to encourage buffer update. Thus, at each iteration $t$, if the norm $\|g_t\|_2$ of the current gradient is larger than the smallest priority key in the buffer, the corresponding critical gradient gets replaced by $g_t$ in the buffer. A standard choice of aggregation function adds $g_t$ to the average of the critical gradients in the buffer.

**The gradient cancellation problem.** However, combining Adam with critical gradients has its pitfalls. We hypothesize that with CG, while the buffer gradients can promote exploration initially (Figure 1), the parameters remain fixed within sharp regions due to *gradient cancellation*. Gradient cancellation primarily occurs when existing buffer gradients are quickly replaced by high-magnitude gradients when the parameters are near a sharp basin. As a result, the buffer quickly converges to high variance gradients whose mean goes to zero, allowing learning to converge. Intuitively, the parameters bounce back and forth off the sides and bottom of the sharp basin: whenever the parameters try to escape the basin, the new outgoing gradient

gets cancelled by incoming gradients in the buffer. Figure 2 illustrates this phenomenon on a toy surface, by showing the buffer gradients (thin blue lines) and their means (black arrow) as well as the new gradient (green arrow), within sharp basins where Adam+CG gets stuck. Additional plots are found in Appendix A.1.

### 3.1 Critical momenta (CM)

As gradient cancellation hinders the ability of Adam+CG to escape sharp minima, our approach addresses this by leveraging a buffer $\mathbf{m}_c$ of *critical momenta* $m_c$ during training. Like McRae et al. (2022), we use the gradient $l_2$-norm as priority criterion[1]. The buffer is a dictionary of key-value pairs $(\|g_c\|_2, m_c)$ with a factor $\lambda \in (0, 1)$ with which the values are decayed at each iteration. The integration with critical momenta leads to a new algorithm, Adam+CM, defined by moment updates:

$$m_t = \beta_1 m_{t-1} + (1 - \beta_1)g_t, \quad m_t^M = \texttt{aggr}(m_t, \mathbf{m}_c), \tag{4}$$

$$v_t^M = \beta_2 v_{t-1}^M + (1 - \beta_2)\,\texttt{aggr}(m_t, \mathbf{m}_c)^2, \tag{5}$$

where $\texttt{aggr}$ is the addition of the current momentum to the average of all critical momenta:

$$\texttt{aggr}(m_t, \mathbf{m}_c) = m_t + \frac{1}{C}\sum_{m_c \in \mathbf{m}_c} m_c. \tag{6}$$

Finally, the Adam+CM update rule is given by

$$\hat{m}_t^M = \frac{m_t^M}{1 - \beta_1^t}, \quad \hat{v}_t^M = \frac{v_t^M}{1 - \beta_2^t}, \quad \theta_{t+1} = \theta_t - \alpha\frac{\hat{m}_t^M}{\sqrt{\hat{v}_t^M + \epsilon}}. \tag{7}$$

The pseudo-code of Adam+CM is given in Algorithm 1.

While at a sharp minimum, the elements of the buffer will still be quickly replaced (Figure 1), due to the inertia in the momentum terms the variance will stay low. Moreover, the fact that gradients quickly change direction will lead to the new momentum terms being smaller and hence have a smaller immediate influence on the aggregate value of the buffer. This allows the overshooting effect to still happen, enabling the exploration effect and helping to learn to escape sharp minima. Furthermore, the larger the size of the buffer, the stronger the overshooting effect will be, and the wider the minimum needs to be for learning to converge. That is because learning needs to stay long enough in the basin of a minimum to fill up most of the buffer in order to turn back to the minimum that it jumped over and for the optimizer to converge. We observe this empirically in Figure 9 and Appendix A.2.2.

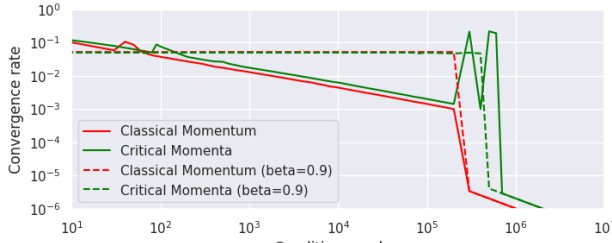

### 3.2 Convergence analysis

In this section, we discuss the convergence of gradient-based algorithms with critical momenta under simplifying assumptions. Leaving aside the specificity of Adam, we consider the simplest variant characterized by the following update,

$$\theta_{t+1} = \theta_t - \alpha\,\texttt{aggr}(m_t, \mathbf{m}_c). \tag{8}$$

In what follows, we restrict our attention to *quadratic losses* and assume that the optimum lies at $\theta^* = 0$ for simplicity, so that $L(\theta) = \frac{1}{2}\theta^\top H\theta$. We also make the following assumptions: $(i)$ as in McRae et al. (2022), we assume a fixed bound on the staleness of momenta, i.e., there is an integer $K > 0$ such that

Figure 3: Quadratic convergence rates $(1 - \rho^*)$ of classical momentum and critical momenta. Solid curves indicate that both $\alpha$ and $\beta$ were optimized to obtain $\rho^*$, while dashed lines indicate that $\rho^*$ obtained with $\beta = 0.9$. Critical momenta converges for a wide range of condition numbers in both cases.

---

[1]We do not use the alternative $\|m_t\|_2$ since the buffer will not get updated fast enough using this criterion.

at each iteration $t$, $m_{t-k} \in \mathbf{m}_C$ implies $k \leq K$; $(ii)$ we further assume that $K$ coincides with the buffer size and that at iteration $t$, $\mathbf{m}_C$ contains *exactly* the momenta $m_{t-1}, \cdots m_{t-C}$.

Under these assumptions, the convergence can be analyzed using standard spectral analysis for multistep linear systems. Considering $V_{t+1} = [\theta_{t+1}, \theta_t, m_t, m_{t-1}, ..., m_{t-C+1}]$, it is straightforward to show that the dynamics can be cast as a linear dynamical system $V_{t+1} = A V_t$ where the matrix $A$ depends on the Hessian $H$, the learning rate $\alpha$ and the momentum parameter $\beta$, and takes the form:

$$
A = \begin{bmatrix}
I - \alpha H & 0 & -\alpha(\beta + \frac{1}{C})I & -\frac{\alpha}{C}I & ... & -\frac{\alpha}{C}I & \frac{\alpha}{C}I \\
I & 0 & 0 & 0 & ... & 0 & 0 \\
H & 0 & \beta I & 0 & ... & 0 & 0 \\
0 & 0 & I & 0 & ... & 0 & 0 \\
\vdots & \vdots & \vdots & \ddots & & \vdots & \vdots \\
0 & 0 & 0 & 0 & ... & I & 0
\end{bmatrix} .
$$

Assuming $L$-smoothness and $\mu$-strongly convexity, the worst-case bound on the convergence rate is obtained by maximizing the spectral radius (largest singular value) $\rho(A)$ over all admissible $H$ (Lessard et al., 2016). Let $h$ denote the eigenvalues of the Hessian matrix $H$. Note that the singular values of the block matrix $A$ are the same as those of the matrices $A_h$, obtained by collapsing the blocks of $A$ by replacing $I$ by 1 and $H$ by $h$. This leads to the simplified problem:

$$
\rho(\alpha, \beta) = \max_{h \in [\mu, L]} \rho(A_h) .
$$

The optimal convergence rate is obtained by tuning these parameters to minimize the spectral radius, i.e., $\rho^* := \min_{\alpha, \beta} \rho(\alpha, \beta)$. Similar to Zhang et al. (2019), we solve this problem numerically using standard solvers to compute the eigenvalues and minimize the worse case spectral radius; we also compare with the convergence rate of critical momenta with classical momentum on diagonal quadratic functions in Figure 3. We plot the convergence rates of classical momentum and critical momenta with $C = 5$ on a quadratic loss function for two variants: (i) Optimal $\alpha, \beta$ (solid curves), (ii) optimal $\alpha$ with fixed $\beta = 0.9$. In both variants, we can see that critical momenta converge for a wide range of condition numbers.

Following the literature (Kaur et al., 2023), we use the maximum eigenvalue ($h_{max}$) of the Hessian $H$ as the indicator of sharpness in the rest of the paper: as $h_{max}$ increases, the surface becomes sharper.

## 4 Insights from toy examples

In this section, we use toy tasks to empirically validate our hypothesis by analyzing and comparing various combinations of Adam with memory augmentation and sharpness-aware minimization.

**Critical momenta promote exploration.** We first compare the optimization trajectories of Adam+CM with Adam, Adam+SAM, and Adam+CG, on interpretable, non-convex 2D loss surfaces. We also include the double combination of Adam with SAM and CM. To complement the Goldstein-Price function in Figure 1, we consider the Ackley function (Ackley, 1987) (see equation 11 in Appendix A.2.1 for the explicit formula), which contains a nearly flat outer region with many sharp minima and a large hole at the center with the global minimum at $(0, 0)$.

We minimize the Ackley function for 10 different initialisation seeds and compare the trajectories of the different optimizers. We run each model for 500 steps and reduce the learning rate by a factor of 10 at the $250^{th}$ step. To get the best-performing setup, we perform a grid search over the hyper-parameters for each optimizer. Figure 4 shows the training curves (left) and optimization trajectories (right) of the different optimizers, for the same initialisation (black square). We observe that, here, only the CM variants are able to explore the loss surface, resulting in a lower loss solution. Additional trajectories with various different seeds for both the Ackley and Goldstein-Price loss surfaces are shown in Appendix A.2.1 (Figure 14 and Figure 13).

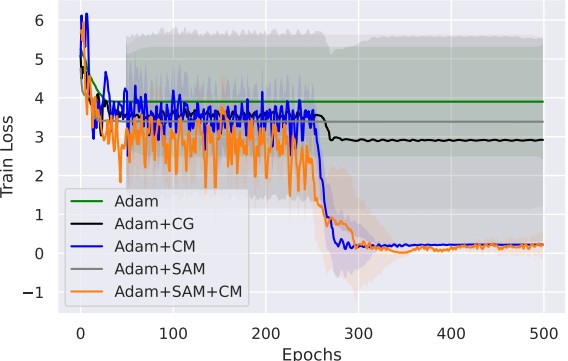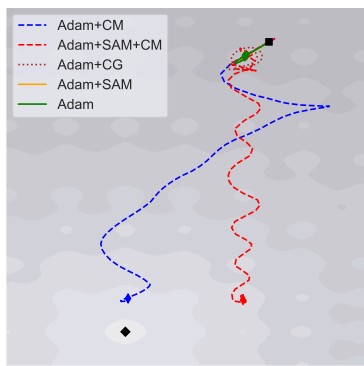

Figure 4: Training loss curves (left, averaged across 10 seeds) and learning trajectories (right, one seed) for different optimizers on the Ackley loss surface. While the other optimizers get stuck in sub-optimal minima near the initialisation point (black square), both CM variants explore and find the lower loss surface near the global solution (black diamond).

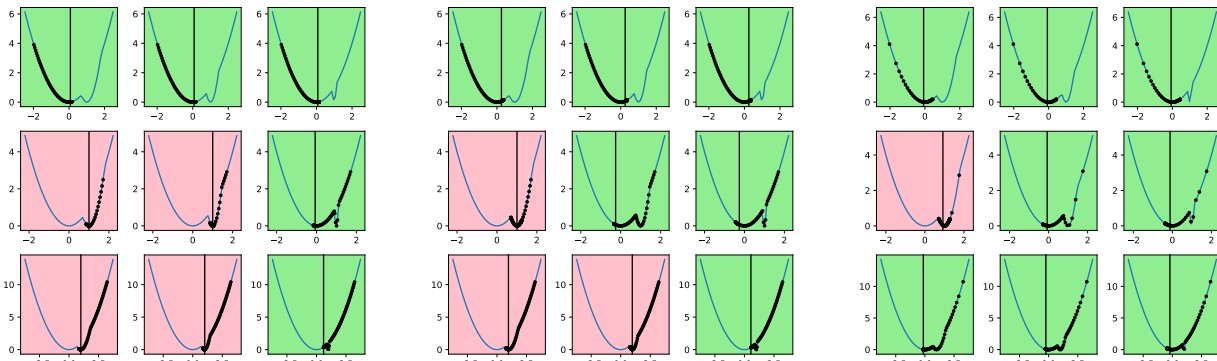

Figure 5: Optimization trajectory of Adam (left), Adam+CG (middle), and Adam+CM (right) on a toy 1D function with a flat and a sharp minimum with increasing sharpness (across columns), for different initialisation points (across rows). Green backgrounds indicate that the optimizer escapes the sharper minimum while red backgrounds indicate otherwise. The vertical line indicates the final point in each sub-figure. We observe that Adam mostly converges to the minimum closest to the initial point. Adam+CM converges to the flatter minimum for different initial points and degrees of sharpness more often than Adam+CG.

**Critical momenta reduce sharpness.** We now want to compare more specifically the implicit bias of the different optimizers towards flat regions of the loss landscape. We first examine the solutions of optimizers trained on the Goldstein-Price and Levy functions (Laguna & Marti, 2005) (see Appendix A.2.1). Both of these functions contain several local minima and one global minimum. We evaluate the solutions based on the final loss and sharpness, averaged across 20 seeds. As a simple proxy for sharpness, we compute the highest eigenvalue of the loss Hessian.

Results in Table 1 show that Adam+CM finds flatter solutions with a lower loss value compared to Adam, Adam+CG, and Adam+SAM in both examples. Furthermore, Adam and Adam+SAM reach almost equal loss values for the Levy function with a negligible difference in sharpness, but for the GP function, Adam+SAM converges to a sub-optimal minimum with lower sharpness.

We analyze how the buffer size controls the amount of exploration empirically in Appendix A.2.1, where we show that even with a small buffer size, Adam+CM can escape sharper minima and explore lower loss regions than other optimizers. The results also suggest that in a controlled setting, the larger buffer size helps find a flatter minimum. To further investigate the escaping abilities of the various optimizers, we consider the

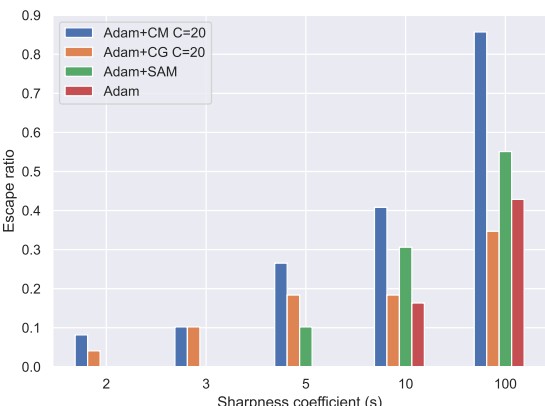

|       | Optimizers | Loss      | Sharpness |
|-------|-----------|-----------|-----------|
|       | Adam      | 0.86      | 1.49      |
| GP    | Adam+SAM  | 3.14      | 1.43      |
|       | Adam+CG   | 0.85      | 1.51      |
|       | Adam+CM   | **0.81**  | **1.36**  |
|       | Adam      | 13.87     | 65.65     |
| Levy  | Adam+SAM  | 13.87     | 65.62     |
|       | Adam+CG   | 13.61     | 64.45     |
|       | Adam+CM   | **12.50** | **62.53** |

Table 1: Loss vs sharpness of the solutions of different optimizers for toy loss surfaces. The buffer decay is set to 0.99 for these experiments. Adam+CM is able to find solutions that are both flatter and deeper (lower loss) than other optimizers in this setting.

Figure 6: Escape ratio (number of times the optimizer escapes the sharp minimum to reach the global minimum out of 50 runs) in the 10-D toy example (equation 9), for different values of the sharpness coefficient. Adam+CM shows a higher ability to escape sharp minima in this setting.

following class of functions on $\mathbb{R}^D$:

$$f_s(x) = \sum_{d=1}^{D} \min(x_d^2, s(x_d - 1)^2) \; , \tag{9}$$

where $s > 1$ is a sharpness coefficient. Each function in this class has two global minima: a flat minimum at the origin and a sharper minimum at $(1 \cdots 1)$.

Figure 5 shows optimization trajectories in the one-dimensional case for various values of the sharpness coefficient $s \in \{5, 10, 100\}$ (across columns) and initial point $x \in \{-2, 2, 3\}$ (across rows). We can see that Adam mostly converges to the minimum closest to the initial point. Adam+CM converges to the flatter minimum for different initial points and degrees of sharpness more often than Adam+CG. Additional plots are shown in Appendix A.3 for various values of the hyperparameters.

In Figure 7 (left), we compare the escape ratio for different optimizers on Equation 9 with $d = 1$ (similar to Figure 6). We can see that for $\mathtt{C} = 10$, with an exception at $s = 2$, Adam+CM consistently finds the minimum at $x = 0$ more often than other optimizers. Interestingly, as $s$ increases, Adam+CG is outperformed by both Adam and Adam+SAM, which indicates that Adam+CG is more susceptible to getting stuck at sharper minima in this example. Figure 7 (right) shows that the tendency of Adam+CM to escape minima is dependent on $\mathtt{C}$ such that, a larger value of $\mathtt{C}$ results in convergence to flatter minimum more often.

Analyzing with $D = 10$, we uniformly sample 50 unique initial points in $[-5, 5]^{10}$. Of these runs, we count the number of times an optimizer finds the flat minimum at the origin by escaping the sharper minimum. Figure 6 reports the escape ratio for different values of the sharpness coefficient. We observe that Adam+CM (with buffer capacity $C = 20$) has a higher escape ratio than others as the sharpness increases. We replicate this experiment with various values of the buffer capacity (see Figure 7).

## 5 Experimental results

Results from the previous section show that our proposed approach of exploration in adaptive optimizers (Adam+CM) is less sensitive to the sharpness of the loss surface, less prone to gradient cancellation, and finds flatter solutions on a variety of loss surfaces. The goal of this section is to evaluate our method empirically on complex models and with the following benchmarks:

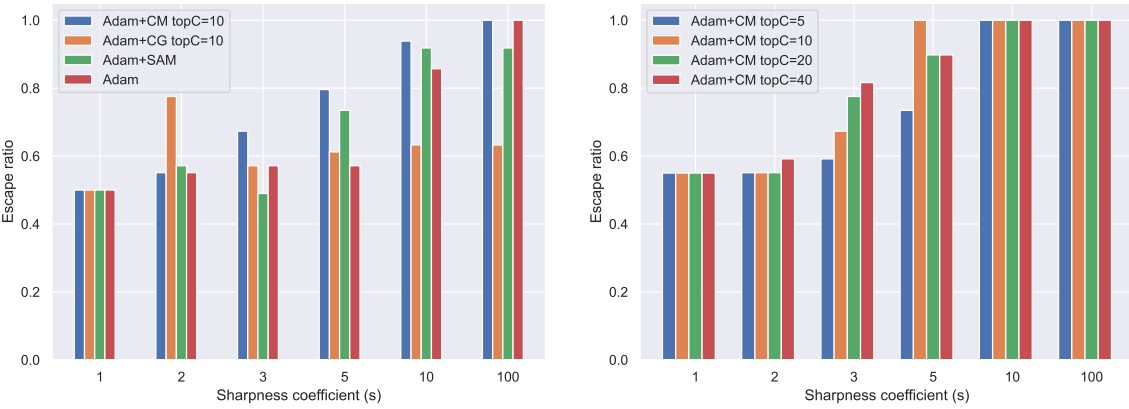

Figure 7: (Left) Escape ratio (number of times when optimizer reaches the minima at $x = 0$ out of total 50 runs) for different sharpness coefficient ($s$) for minima at $x = 1$ in 1D toy example shown in Figure 5. (Right) Escape ratio vs sharpness coefficient ($s$) for different $C$.

- The Penn Treebank (PTB) (Marcus et al., 1993): It is a part-of-speech (POS) tagging task where a model must determine what part of speech (ex. verb, subject, direct object, etc.) every element of a given natural language sentence consists of.

- CIFAR10 (Krizhevsky et al., 2009): It is a multiclass image classification task the training and validation set consists of images that are separated into 10 distinct classes, with each class containing an equal number of samples across the training and validation sets. The goal is to predict the correct image class, provided as an annotated label, given a sample from the sets.

- CIFAR100: It is the same task as CIFAR10, except that images are now separated into 100 classes with an equal number of samples within each class.

- ImageNet (Deng et al., 2009): It is a large-scale dataset consisting of images separated into 1000 distinct classes. The objective of the task is the same as CIFAR10 and CIFAR100, which is to classify images from the training and validation set correctly into a class with an annotated label.

All results presented in this section are averaged across five seeds.

## 5.1 Language modelling

Starting with a language-based task, a single-layer long short-term memory network (LSTM) (Hochreiter & Schmidhuber, 1997) is trained on the PTB dataset. We evaluate the performance by reporting the validation perplexity on a held-out set. We train the model for 50 epochs (similar to McRae et al. (2022)) and we reduce the learning at the $25^{th}$ epoch by dividing it by 10. The results are reported after performing a grid search over corresponding hyper-parameters. Details of this grid search are present in Appendix Table 8. Table 2 shows that Adam+CM outperforms other optimizers in terms of best validation perplexity achieved during training. We also note that Adam+SAM is the second-best optimizer and achieves better performance than its CM variant.

## 5.2 Image classification

Next, we evaluate Adam+CM on different model sizes for image classification. We train ResNet34 models (He et al., 2016) on CIFAR10/100 datasets. We train the models for 100 epochs where we reduce the learning at the $50^{th}$ epoch by dividing it by 10. We also train an EfficientNet-B0 model (Tan & Le, 2019) from scratch on ImageNet. We used a publicly available EfficientNet implementation[2] in PyTorch (Paszke et al., 2019),

---

[2]https://github.com/lukemelas/EfficientNet-PyTorch

| Optimizers | Language modelling Validation Perplexity ($\downarrow$) | Image classification Validation Accuracy % ($\uparrow$) | | |
|---|---|---|---|---|
| | PTB | CIFAR10 | CIFAR100 | ImageNet |
| Adam | $179.4_{\pm 2.8}$ | $93.9_{\pm 0.3}$ | $70.7_{\pm 0.3}$ | $67.8_{\pm 0.1}$ |
| Adam+CG | $174.4_{\pm 5.5}$ | $93.8_{\pm 0.4}$ | $71.0_{\pm 0.3}$ | $69.7_{\pm 0.1}$ |
| Adam+SAM | $168.7_{\pm 1.9}$ | $93.7_{\pm 0.3}$ | $70.5_{\pm 0.4}$ | $65.4_{\pm 0.2}$ |
| Adam+CM (ours) | $\mathbf{163.2_{\pm 5.5}}$ | $94.0_{\pm 0.3}$ | $\mathbf{71.2_{\pm 0.3}}$ | $\mathbf{71.7_{\pm 0.2}}$ |
| Adam+SAM+CM (ours) | $176.2_{\pm 5.7}$ | $\mathbf{94.4_{\pm 0.4}}$ | $69.7_{\pm 0.3}$ | $71.3_{\pm 0.2}$ |

Table 2: Comparison of performance in terms of best validation perplexity and accuracy (%) achieved by the existing baselines with Adam+CM and its SAM variant on language modelling and image classification tasks. Overall, CM outperforms the baselines in all four datasets.

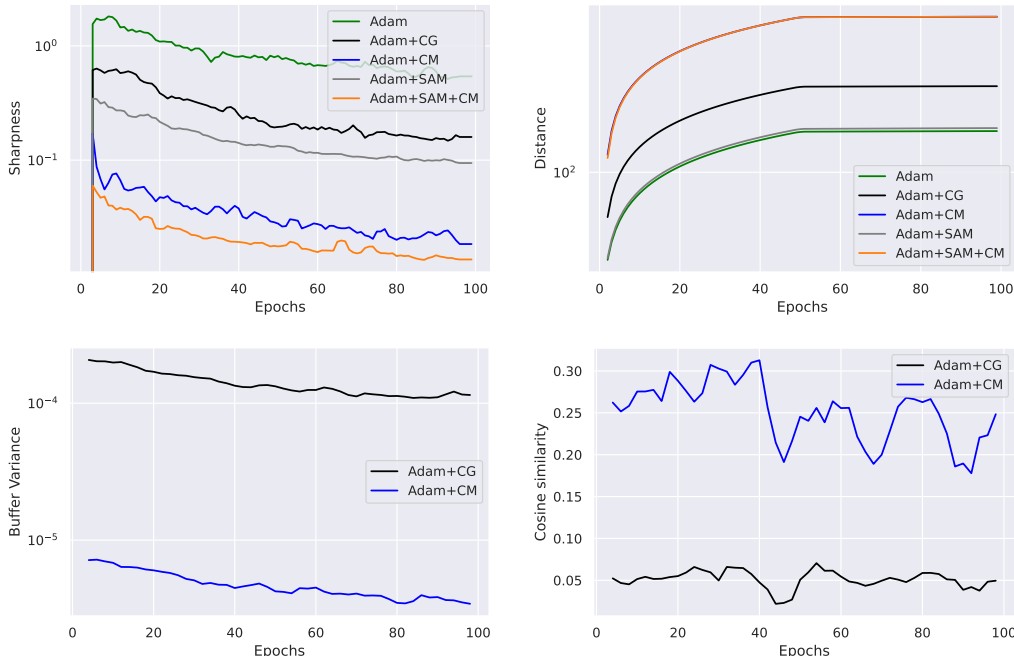

Figure 8: Sharpness (top-left), distance (top-right) buffer variance (bottom-left), and cosine similarity (bottom-right) in buffer elements of the optimizers on CIFAR100. These results indicate that buffer elements in Adam+CM agree more with each other and have lower sharpness than Adam+CG.

with a weight decay (Loshchilov & Hutter, 2019) of $10^{-4}$ and a learning rate scheduler where the initial learning rate is reduced by a factor of 10 every 30 epochs. We provide additional details about the grid search, datasets, and models in Appendix A.2.2.

The best validation accuracy achieved by models for image classification tasks is reported in Table 2. We again observe that Adam+CM achieves better performance than the other optimizer baselines. Moreover, Adam+SAM+CM yielded the best validation accuracy for CIFAR10, while Adam+CM performed the best on the other two datasets.

Figure 8 corroborates the claim that Adam+CM finds a flatter surface containing the global minimum, with the top-left plot showing lower sharpness compared to Adam or Adam+SAM. The top-right plot further reveals the greater distance travelled by parameters during training, indicating that using CM promotes more exploration than the other optimizers. We also note that the distance travelled by parameters in SAM is similar to that of Adam. This similarity is also observed with Adam+CM and Adam+SAM+CM as their

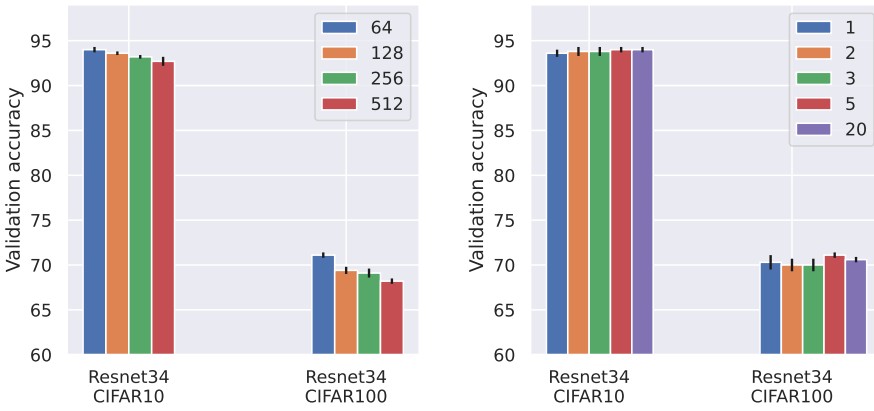

Figure 10: Best validation accuracy for different batch sizes (left) and buffer size (right) using Adam+CM on CIFAR10/100, with hyperparameters grid search. The default batch size of 64 and buffer size of 5 works best for these benchmarks.

distance curves overlap. Figure 8 (bottom-left) shows that buffer elements stored by Adam+CM have lower variance during training compared to Adam+CG. To compare agreement among buffer quantities, we take the element with the highest norm within the buffer, compute the cosine similarities with other elements in the buffer, and take the mean of these similarities.

Figure 8 (bottom-right) shows the agreement in Adam+CM remains higher than in Adam+CG, indicating the aggregation of buffer elements in Adam+CM will more often result in a non-zero quantity in the desired direction. On the other hand, high variance and disagreement among elements in the Adam+CG buffer may cause gradient cancellation during aggregation and result in Adam-like behavior. These results are associated with the best-performing setup for the given optimizers. They highlight that it is the property of Adam+CM that facilitates efficient exploration and finding flatter surfaces to achieve the best performance.

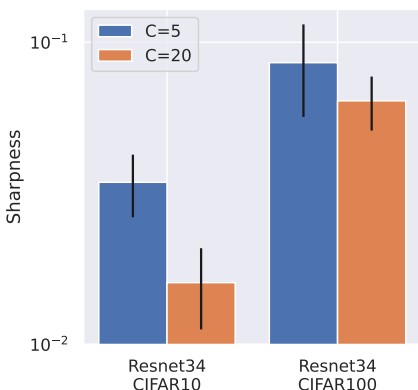

Figure 9 shows the final sharpness metric for different buffer sizes recorded for CIFAR10/100 experiments with default hyperparameter setup. It is clear that using a large buffer size can further reduce the sharpness of the solution in such complex settings.

Next, we show how batch size influences the performance of Adam+CM for CIFAR10/100 experiments. In both datasets, we found that the default batch size of 64 performs best in terms of validation accuracy. We provide the validation accuracy achieved by the best-performing setups for other batch sizes in Figure 10 (left). We also perform a similar experiment but for different buffer sizes on CIFAR10/100 experiments. We found that the default buffer size of 5 performs best in terms of validation accuracy. We provide the

Figure 9: Sharpness for different buffer sizes using Adam+CM on CIFAR10/100, with other hyperparameters fixed. Using larger buffers results in lower sharpness even for high-dimensional models.

best-performing setups in Figure 10 (right) and also observe that smaller buffer size results have a slightly higher standard deviation.

## 5.3 Using non-adaptive optimizer

Next, we show the advantage of integrating exploration capabilities in SGD. To implement SGD+CG, we follow McRae et al. (2022) and replace $g_t$ with aggregation of $g_t$ and the buffer gradients $\mathbf{g}_c$ for parameter

| Optimizers | CIFAR10 | | CIFAR100 | |
|---|---|---|---|---|
| | Accuracy | Speed-up | Accuracy | Speed-up |
| SGD | $93.0_{\pm 0.4}$ | 1x | $69.6_{\pm 0.7}$ | 1x |
| SGD+CG | $93.1_{\pm 0.4}$ | 1.2x | $69.6_{\pm 0.3}$ | 1.2x |
| SGD+CM (ours) | $93.1_{\pm 0.1}$ | 1.3x | $69.6_{\pm 0.7}$ | 1.2x |

Table 3: Comparison of performance in terms of best validation accuracy (%) and speed-up by SGD+CM on CIFAR10 and CIFAR100. Although the performance remains the same, there is a speed-up by both memory-augmented optimizers.

updates:

$$\theta_{t+1} = \theta_t - \eta \left( g_t + \frac{1}{C} \sum_{g_c \in \mathbf{g}_c} g_c \right) . \tag{10}$$

Similarly, for SGD+CM, we aggregate the momentum using equation 4 and update the parameters using equation 8. We perform a hyperparameter grid search on CIFAR10/100 experiments to compare SGD, SGD+CG with SGD+CM and report the results in Table 3. Apart from validation accuracy, we also report the speed-up in the first 50 epochs of the training process and observe that although generalization performance remains the same, there is a speed-up in both SGD+CG and SGD+CM.

### 5.4 Online learning

We also evaluate the methods in an online learning setup where a model with a limited capacity is trained on a stream of tasks. Here, the goal is to adapt to the latest task dataset by maximizing its performance. Therefore, it is important for an optimizer to escape task-specific solutions as the loss landscape changes once a new task dataset arrives. We hypothesize that with its greater exploration capabilities, Adam+CM will be able to do so better than other baselines. We evaluate the optimizers on the following benchmarks:

- TinyImagenet (Zhang et al., 2019): This dataset is created by partitioning its 200 classes into 40 5-way classification tasks. The implementation of TinyImagenet is based on Gupta et al. (2020) where a 4-layer CNN model is trained.

- 5-dataset (Mehta et al., 2023): It consists of five different 10-way image classification tasks: CIFAR10, MNIST LeCun (1998), Fashion-MNIST Xiao et al. (2017), SVHN Netzer et al. (2011), and notMNIST Bulatov (2011). The implementation is based on Mehta et al. (2023) where a ResNet18 He et al. (2016) model is trained.

We report their learning accuracies, which is the average validation accuracy for each task when the model is trained on that specific task. More details about the models, datasets, and hyper-parameter grid search are provided in Appendix A.2.3.

In Table 4, we observe that Adam+CM achieves better performance than Adam. Moreover, while Adam+SAM performs similarly to Adam, Adam+SAM+CM results in the best accuracy on both benchmarks. One possible reason for this is that, while SAM primarily focuses on improving the flatness of a solution, CM is more effective in exploring the loss surface and finding a basin containing a more generalizable solution. Our findings validate the hypothesis that CM can be applied in

| Optimizers | TinyImagenet | 5-dataset |
|---|---|---|
| Adam | $62.9_{\pm 0.4}$ | $88.2_{\pm 0.5}$ |
| Adam+CG | $62.8_{\pm 0.1}$ | $88.2_{\pm 0.5}$ |
| Adam+SAM | $62.8_{\pm 0.4}$ | $88.3_{\pm 1.0}$ |
| Adam+CM | $\underline{63.4}_{\pm 0.1}$ | $\underline{88.4}_{\pm 0.6}$ |
| Adam+SAM+CM | $\mathbf{63.9}_{\pm \mathbf{0.7}}$ | $\mathbf{88.5}_{\pm \mathbf{0.5}}$ |

Table 4: Performance comparison in terms of best learning accuracy (%) of the existing baselines with Adam+CM and its SAM variant on online learning benchmarks.

online learning settings to promote exploration. Moreover, it complements existing baselines and can be seamlessly integrated for performance improvements.

## 6    Conclusion

This work presents a framework for enhancing exploration in adaptive optimizers. We introduce Adam+CM, a novel memory-augmented variant of Adam that incorporates a buffer of critical momenta and adapts the parameters update rule using an aggregation function. Our analysis demonstrates that Adam+CM effectively mitigates the limitations of existing memory-augmented adaptive optimizers, fostering exploration toward flatter regions of the loss landscape. Empirical evaluations showcase that Adam+CM outperforms Adam, SAM, and CG across standard supervised and online learning tasks. An exciting future direction is the application of reinforcement learning settings, where knowledge transfer without overfitting on a single task is crucial. Furthermore, our findings hint at the potential of CM to capture higher-order dynamics of the loss surface, warranting further investigation. Even though the quadratic assumption in our theoretical analysis can be (slightly) relaxed by assuming that the loss Hessian matrices computed along the optimization trajectory all commute with each other, analyzing broader classes of functions is worth exploring in the future.

## Acknowledgments

This research was supported by Samsung Electronics Co., Ltd. through a Samsung/Mila collaboration grant, and was enabled in part by compute resources provided by Mila, the Digital Research Alliance of Canada, and NVIDIA. Pranshu Malviya is partially supported by the Merit scholarship program for foreign students (PBEEE) by Fonds de Recherche du Québec Nature et technologies (FRQNT). Gonçalo Mordido was supported by an FRQNT postdoctoral scholarship (PBEEE) during part of this work. Simon Lacoste-Julien is a CIFAR Associate Fellow in the Learning Machines & Brains program and supported by NSERC Discovery Grants. Sarath Chandar is supported by the Canada CIFAR AI Chairs program, the Canada Research Chair in Lifelong Machine Learning, and the NSERC Discovery Grant.

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

# A  Appendix

In this section, we provide the details and results not present in the main content. In section A.1, we report more evidence of gradient cancellation in a toy example. In section A.2, we describe the implementation details including hyper-parameters values used in our experiments. All experiments were executed on an NVIDIA A100 Tensor Core GPUs machine with 40 GB memory.

## A.1  Gradient cancellation in CG

When we track the trajectory along with the gradient directions of the buffer in the Adam+CG optimizer on the Ackley+Rosenbrock function (defined in the next section), we found that CG gets stuck in the sharp minima (see Figure 11). This is because of the gradient cancellation problem discussed earlier.

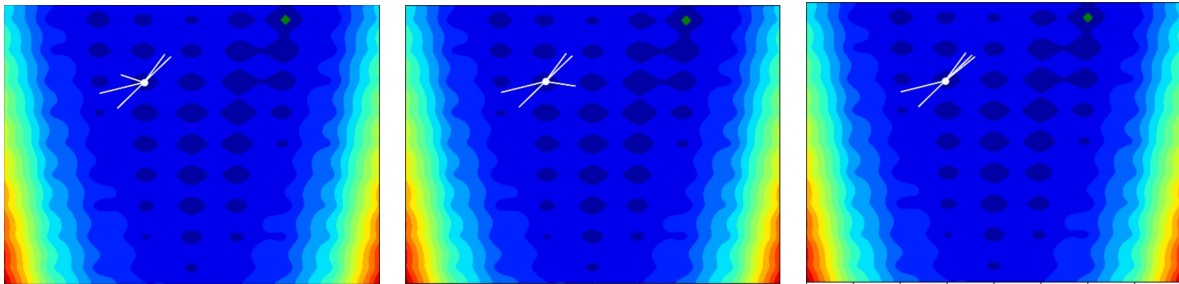

Figure 11: Three consecutive steps in training Adam+CG on Ackley+Rosenbrock function with $C = 5$. The white lines indicate the gradient directions in the buffer. Since four of these gradients point in opposite directions, their mean would be small and cancel the current gradient update out. As a result, parameters get stuck in the sharp minima.

Next, we provide additional analyses similar to Figure 2. We compare the trajectories and gradient directions of Adam+CM and Adam+CG for six consecutive steps on the Goldstein–Price function in Figure 12. Similar to our observations on the Ackley function, we find that the buffer quantities in CG are dominated by directions of higher sharpness (towards the top-left direction and bottom-right direction). Even when the new gradient direction aligns with the buffer mean in CG, it remains stuck in the sharp surface. On the other hand, the buffer directions do not drastically change in CM and therefore it finds the surface that contains the global minima.

## A.2  Implementation details and other results

### A.2.1  Toy examples

We evaluate our optimizer on the following test functions in Section 4:

1. Ackley function:

$$f(x, y) = -20 \exp(-0.2\sqrt{0.5(x^2 + y^2)}) - \exp(0.5(\cos 2\pi x + \cos 2\pi y)) + e + 20 \ . \tag{11}$$

   The global minimum is present at $(0, 0)$. In Figure 13, we visualize the trajectories of Adam, Adam+SAM, Adam+CG, and Adam+CM for different initialisation points. While other optimizers may get stuck at nearby local minima, Adam+CM benefits from more exploration and finds a lower-loss surface that may contain the global minima.

2. Goldstein-Price function:

$$\begin{aligned} f(x, y) = \frac{1}{2.427} &\log[1 + (x + y + 1)^2(19 - 14x + 3x^2 - 14y + 6xy + 3y^2)] \\ &\times [30 + (2x - 3y)^2(18 - 32x + 12x^2 + 48y - 36xy + 27y^2) - 8.693] \ . \end{aligned} \tag{12}$$

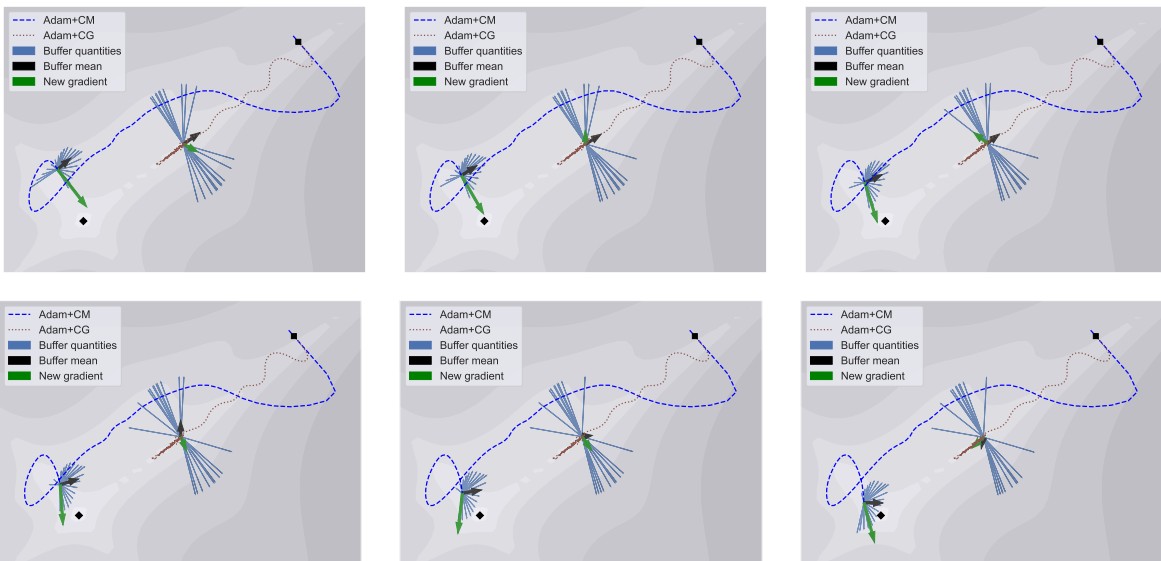

Figure 12: Six consecutive steps of Adam+CG and Adam+CM trajectories on Goldstein–Price function function. Buffer quantities in Adam+CG are dominated by directions of higher sharpness (towards top-left direction and bottom-right direction) and the new gradient directions have high variance, yielding a small update. Conversely, Adam+CM escapes sub-optimal minima and converges near the global minimum.

The global minimum is present at $[0, 1]^2$. In Figure 14, we visualize the trajectories of Adam, Adam+SAM, Adam+CG, and Adam+CM for different initialisation points. While other optimizers may get stuck at a sub-optimal loss surface, Adam+CM benefits from more exploration and finds the global minimum.

3. Levy function:

$$f(x_1, x_2) = \sin^2(\pi w) + \sum_{i=1}^{d-1}(w_i - 1)^2[1 + 10\sin^2(\pi w_i + 1)] + (w_d - 1)^2[1 + \sin^2(2\pi w_d)] . \qquad (13)$$

where $w_i = 1 + \frac{x_i - 1}{4}$ and $d$ is the number of variables. The global minimum is present at $(1, 1)$.

4. Ackley+Rosenbrock function:

$$\begin{aligned} f(x, y) = 0.05(1 - x)^2 + 0.05(y - x^2)^2 + 0.6[\exp(-0.2\sqrt{0.5(x^2 + y^2)}) \\ - \exp(0.5(\cos 2\pi x + \cos 2\pi y)) + e] . \end{aligned} \qquad (14)$$

The global minimum is present at $(2.1, 3.9)$.

In Figure 1, we also compared different optimizers in terms of the sharpness of the solution reached by each optimizer on different functions. In Table 5, we perform a similar experiment and compare sharpness across different values of hyperparameters for Adam+CM. We observe that Adam+CM is able to find a flatter and lower-loss solution as buffer size increases. This is consistent with complex tasks in Figure 9.

We also provide a comparison with another existing method called Momentum Extragradient (MEG) (Kim et al., 2022a) and Adam+CM on the Ackley and Goldstein-Price functions in Figure 15. We observe that Adam+CM has better exploration capabilities than MEG. When evaluated on CIFAR10 dataset, Adam+MEG achieves the accuracy of $93.8_{\pm 0.3}$ % which is lower than both Adam+CM ($94.0_{\pm 0.3}$ %) and Adam+SAM+CM ($94.4_{\pm 0.4}$ %).

In Figure 4, we compared different variants of Adam on Ackley function by running 500 steps. We observed that, unlike other baselines, Adam+CM and Adam+SAM+CM escape local minima, explore the loss surface,

| Optimizers | C | $\lambda$ | Loss | Sharpness |
|------------|-----|------|-------|-----------|
| Adam+CG | 5 | 0.7 | 14.40 | 64.72 |
| Adam+CG | 5 | 0.99 | 14.41 | 64.67 |
| Adam+CG | 20 | 0.99 | 13.61 | 64.45 |
| Adam+CM | 5 | 0.7 | 12.90 | 63.74 |
| Adam+CM | 5 | 0.99 | 13.02 | 63.83 |
| Adam+CM | 20 | 0.99 | **12.50** | **62.53** |

Table 5: Loss vs sharpness of the solutions of Adam+CM for different hyperparameters buffer sizes on Levy function. Adam+CM is able to find solutions that are both flatter and deeper (lower loss) as buffer size increases. Moreover, Adam+CM always outperforms Adam+CG even with a smaller buffer size.

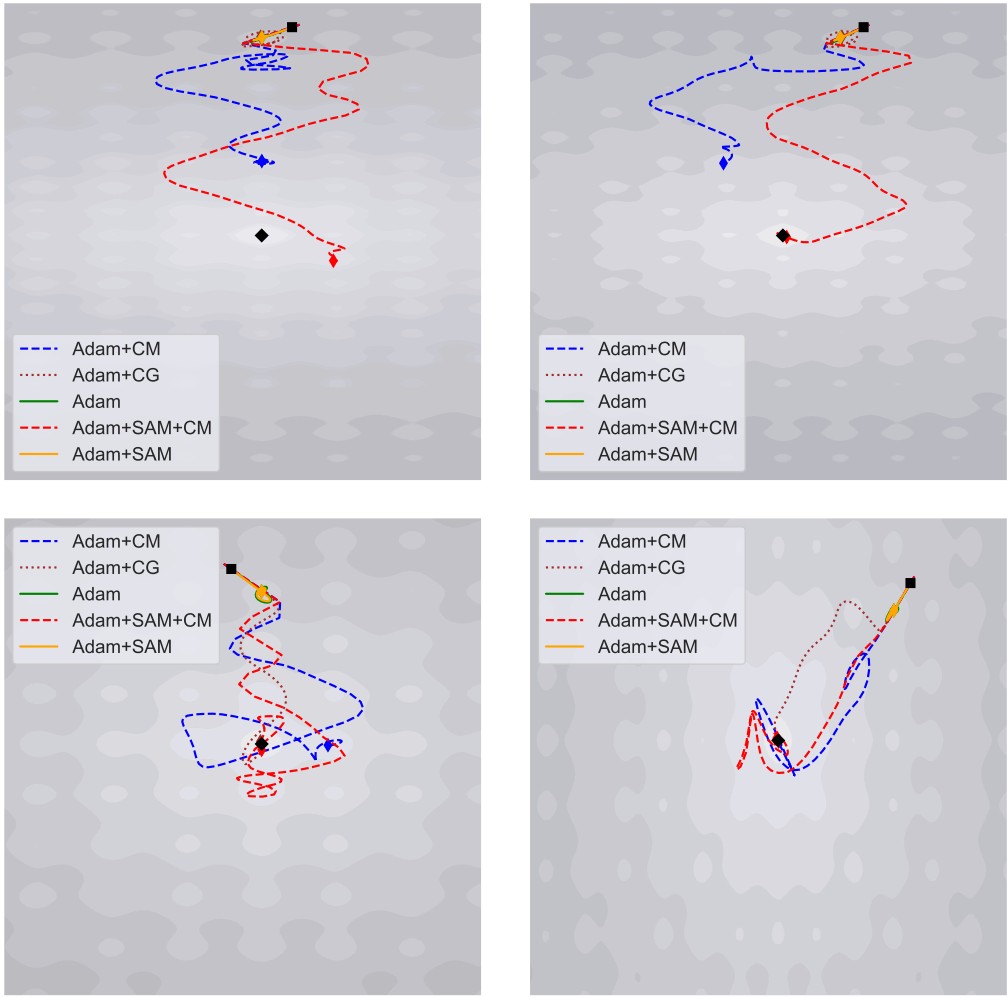

Figure 13: Optimization trajectories of various optimizers on the Ackley loss surface for different initial points (black square). The black diamond indicates the global minimum.

and navigate towards the region containing global minima. In this experiment, reducing the learning rate helps in settling down to the global minimum and stopping further exploration. As an ablation study, we perform the same experiment but without learning rate decay. We plot the loss curve in Figure 16 and observe that Adam+CM oscillates near the global minima.

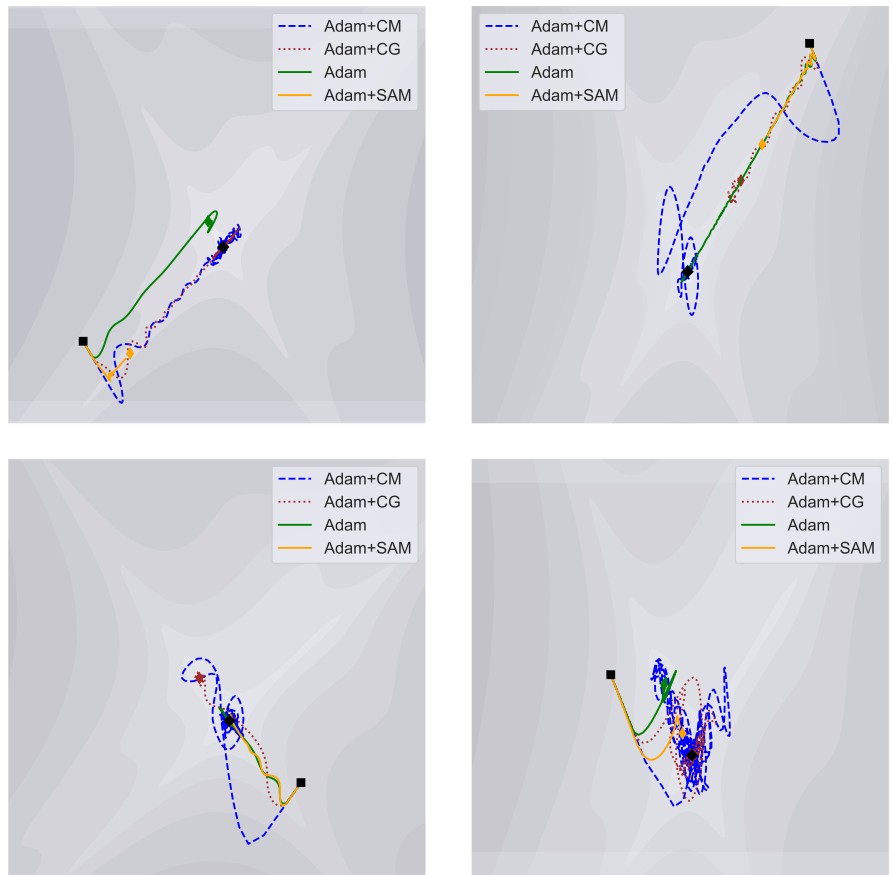

Figure 14: Optimization trajectories of various optimizers on the Goldstein-Price loss surface for different initial points (black square). The black diamond indicates the global minimum..

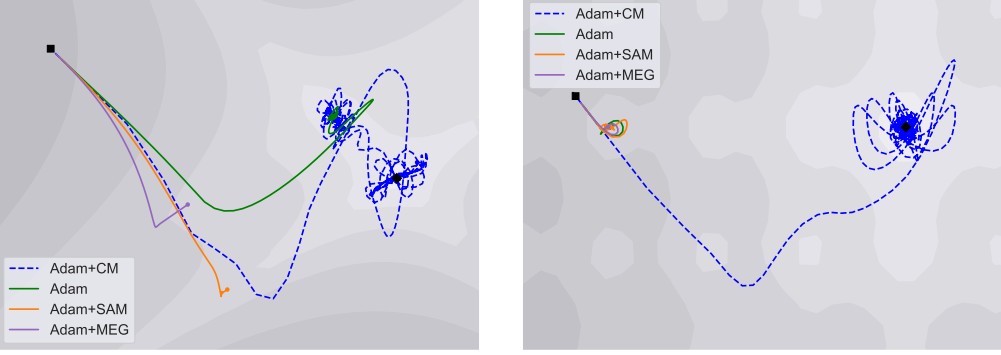

Figure 15: Comparing adaptation trajectories of Adam+MEG with Adam, Adam+SAM and Adam+CM. Overall, we observe that Adam+CM exhibits better exploration of the loss landscape.

### A.2.2 Deep learning experiments

In Table 6 and Table 7, we provide a summary of all datasets and deep learning models used in the experiment from Section 5.

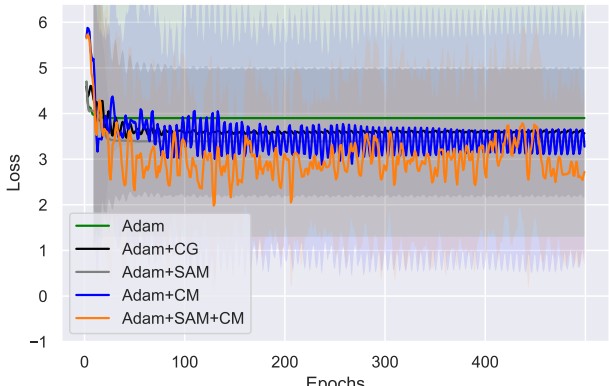

Figure 16: Loss (averaged across 10 seeds) obtained on and learning trajectories for different optimizers with a constant learning rate on the Ackley loss surface. We observe that while other optimizers get stuck at sub-optimal minima, Adam+CM and Adam+SAM+CM achieve lower loss values and oscillates near the global minima.

| Dataset | Train set | Validation set |
|---------|-----------|----------------|
| PTB | 890K | 70K |
| CIFAR10 | 40K | 10K |
| CIFAR100 | 40K | 10K |
| ImageNet | 1281K | 50K |

Table 6: Dataset details

| Model | Number of parameters |
|-------|----------------------|
| LSTM | 20K |
| ResNet34 | 22M |
| EfficientNet-B0 | 5.3M |

Table 7: Model details

For all toy examples experiments based on Equation 9, the learning rate is set as 0.05. The learning rate is set as 0.1 for other toy examples. Unless specified in the experiment description, the default set of other hyperparameters in all our experiments is $\{\beta_1, \beta_2, \mathtt{C}, \lambda, \rho\} = \{0.9, 0.99, 5, 0.7, 0.05\}$ except in CIFAR10/100 experiments where $\beta_2$ is set to 0.999. The default values of $\mathtt{C}$ and $\lambda$ are decided based on the suggested values from McRae et al. (2022) and $\rho$ based on Foret et al. (2021). For the results in Figure 6 and Figure 1, $\mathtt{C}$ and $\lambda$ are set to 20 and 0.99.

For supervised learning experiments, we provide the details on hyper-parameter grid-search in Table 8 and the best settings for all experiments and Table 9 and Table 10. In these tables, we report the hyperparameter set for each optimizer as follows:

- Adam: $\{\alpha\}$

- Adam+CG: $\{\alpha, \beta_1, \beta_2, \mathtt{C}, \lambda\}$

- Adam+SAM: $\{\alpha, \beta_1, \beta_2, \rho\}$

- Adam+CM: $\{\alpha, \beta_1, \beta_2, \mathtt{C}, \lambda\}$

- Adam+SAM+CM: $\{\alpha, \beta_1, \beta_2, \mathtt{C}, \lambda, \rho\}$

CM essentially uses the equivalent cost of computation and space complexity as CG. However, after performing a hyperparameter grid search over the buffer size, we observed that CM consistently requires a smaller buffer size than CG and obtains better performance with ($C = 5, \lambda = 0.7$).

| Hyper-parameter | Set |
|---|---|
| lr | $\{0.1, 0.01, 0.001, 0.0001\}$ |
| $\beta_1$ | $\{0.9, 0.99, 0.999\}$ |
| $\beta_2$ | $\{0.99, 0.999, 0.9999\}$ |
| C | $\{5, 20\}$ |
| $\lambda$ | $\{0.7, 0.99\}$ |
| $\rho$ | $\{0.01, 0.05, 0.1\}$ |

Table 8: Details on grid search on hyper-parameter setting.

| Optimizers | PTB LSTM |
|---|---|
| Adam | $\{0.001, 0.9, 0.99\}$ |
| Adam+CG | $\{0.001, 0.9, 0.999, 5, 0.7\}$ |
| Adam+SAM | $\{0.001, 0.9, 0.9, 0.01\}$ |
| Adam+CM | $\{0.001, 0.9, 0.999, 5, 0.7\}$ |
| Adam+SAM+CM | $\{0.001, 0.9, 0.999, 5, 0.7, 0.1\}$ |

Table 9: Best hyperparameter settings for different optimizers on PTB.

| Optimizers | CIFAR10 | CIFAR100 | ImageNet |
|---|---|---|---|
| Adam | $\{0.001, 0.9, 0.999\}$ | $\{0.001, 0.9, 0.99\}$ | $\{0.0001, 0.9, 0.99\}$ |
| Adam+CG | $\{0.0001, 0.9, 0.999, 20, 0.7\}$ | $\{0.001, 0.9, 0.99, 20, 0.7\}$ | $\{0.0001, 0.9, 0.99, 5, 0.7\}$ |
| Adam+SAM | $\{0.0001, 0.9, 0.99, 0.05\}$ | $\{0.001, 0.9, 0.999, 0.05\}$ | $\{0.0001, 0.9, 0.99, 0.05\}$ |
| Adam+CM | $\{0.0001, 0.9, 0.999, 5, 0.7\}$ | $\{0.001, 0.9, 0.9999, 5, 0.7\}$ | $\{0.0001, 0.9, 0.99, 5, 0.7\}$ |
| Adam+SAM+CM | $\{0.0001, 0.9, 0.99, 5, 0.7, 0.05\}$ | $\{0.001, 0.9, 0.99, 5, 0.7, 0.1\}$ | $\{0.0001, 0.9, 0.99, 5, 0.7, 0.05\}$ |

Table 10: Best hyperparameter settings for different optimizers on image classification benchmarks.

### A.2.3 Online learning setup

The online learning experiments are performed on the following benchmarks:

- **TinyImagenet**: This dataset is created by partitioning its 200 classes into 40 5-way classification tasks. The implementation of TinyImagenet is based on Gupta et al. (2020) where a 4-layer CNN model is trained.

- **5-dataset**: It consists of five different 10-way image classification tasks: CIFAR10, MNIST LeCun (1998), Fashion-MNIST Xiao et al. (2017), SVHN Netzer et al. (2011), and notMNIST Bulatov (2011). The implementation is based on Mehta et al. (2023) where a ResNet18 He et al. (2016) model is trained.

For online learning experiments, we provide the details on hyper-parameter grid-search in Table 12 and the best settings for all experiments in Table 13.

### A.3 Sensitivity analysis

Following experiments in Figure 5, we fix decay to 0.7 in Figure 17 and vary C. We perform a similar experiment with decay= 0.99 and plot them in Figure 18. In both these figures, the observation remains

| Optimizers | GP | Ackley |
|---|---|---|
| Adam | {0.1, 0.9, 0.99} | {0.1, 0.9, 0.99} |
| Adam+CG | {0.1, 0.9, 0.99, 5, 0.7} | {0.1, 0.9, 0.99, 5, 0.7} |
| Adam+SAM | {0.1, 0.9, 0.99, 0.01} | {0.1, 0.9, 0.99, 0.01} |
| Adam+CM | {0.1, 0.9, 0.999, 20, 0.99} | {0.1, 0.9, 0.999, 20, 0.99} |
| Adam+SAM+CM | {0.1, 0.9, 0.999, 20, 0.99, 0.05} | {0.1, 0.9, 0.999, 20, 0.99, 0.05} |

Table 11: Best hyperparameter settings for different optimizers on toy examples.

| Hyper-parameters | Values |
|---|---|
| step size ($\eta$) | $\{0.01, 0.001, 0.0001, 0.00001\}$ |
| $\beta_1$ | $\{0.9, 0.99\}$ |
| $\beta_2$ | $\{0.99, 0.999, 0.9999\}$ |

Table 12: Details on the hyper-parameter grid search used for the online learning experiments.

| Optimizer | TinyImagenet | 5-dataset |
|---|---|---|
| Adam | {0.0001, 0.9, 0.999} | {0.0001, 0.9, 0.999} |
| Adam+CG | {0.0001, 0.9, 0.999} | {0.0001, 0.9, 0.999} |
| Adam+SAM | {0.0001, 0.9, 0.9999} | {0.001, 0.9, 0.999} |
| Adam+CM | {0.0001, 0.9, 0.9999} | {0.001, 0.9, 0.9999} |
| Adam+SAM+CM | {0.0001, 0.9, 0.99} | {0.00001, 0.9, 0.999} |

Table 13: Best hyper-parameter settings for online learning experiments.

the same that is Adam+CM converges to flatter minima for different initial points and degrees of sharpness. We also observe that C plays an important role in convergence to flatter minima in both Adam+CG and Adam+CM.

### A.3.1   $m$-**sharpness**

There have been various definitions of sharpness used in optimization literature. Specifically, Foret et al. (2021) employs $m$-sharpness to indicate the sharpness of the loss landscape and demonstrates its correlation with generalization performance.

The $m$-sharpness is defined as:

$$\frac{1}{n} \sum_{M \in D} \max_{\|\epsilon\|_2 \leq r} \frac{1}{m} \sum_{s \in M} L_s(\theta + \epsilon) - L_s(\theta) \ , \tag{15}$$

where $D$ represents the training dataset, which is composed of $n$ mini-batches $M$ of size $m$ and $r$ is a hyper-parameter set to 0.05 by default.

In Figure 19, we also monitor the $m$-sharpness during the training of CIFAR10 (left) and CIFAR100 (right) with the same learning rate of 0.0001 and fixed hyperparameter setups. We compare the baseline optimizers with Adam+CM and observe that Adam+CM exhibits lower $m$-sharpness on both datasets, which is consistent with our observations in Figure 8.

### A.3.2   Learning rate and sharpness

In this section, we compare generalization performances and empirical sharpness of the solutions obtained using the Adam optimizer with different learning rates on CIFAR10. We keep the other hyper-parameters fixed to their default values for fair comparison. We show that even when increasing the learning rate decreases the overall sharpness (Figure 20), the resulting minima is sub-optimal (Figure 14).

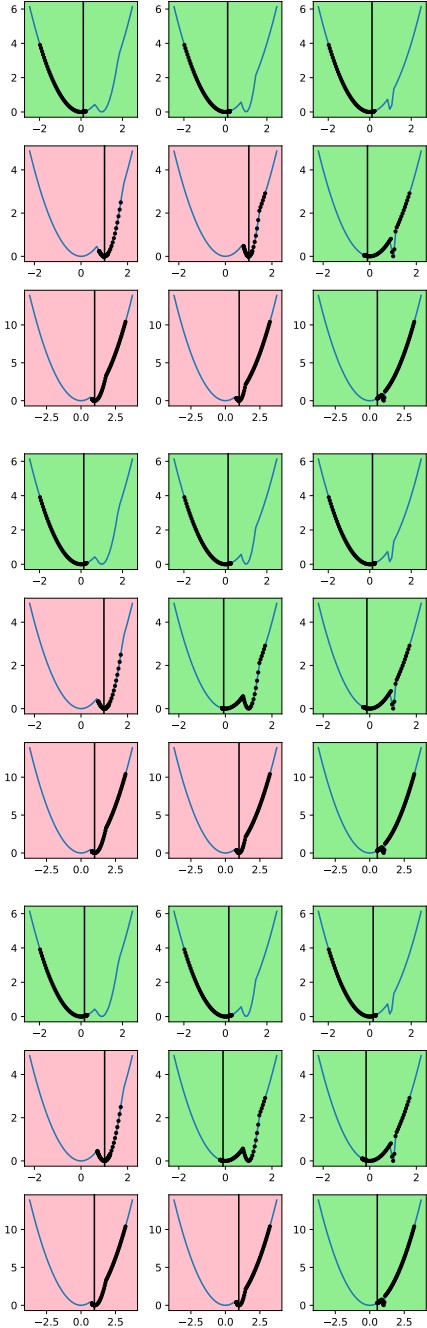
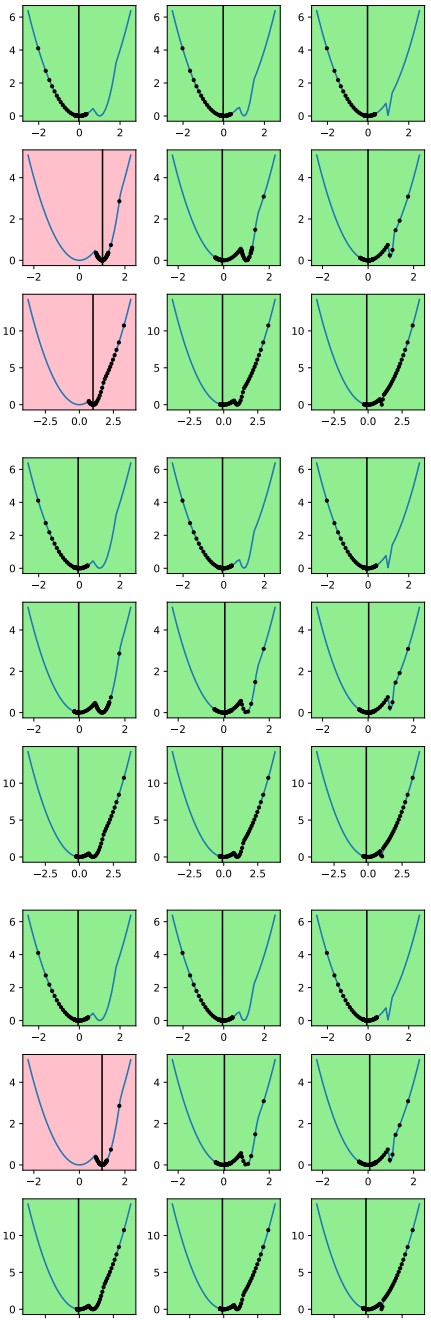

Figure 17: Following Figure 5, we compare trajectories of Adam+CG (left columns) and Adam+CM (right column) on Equation 9 with $d = 1$ where $\lambda$ is set to 0.7 and $C$: (i) 5 (first row), (ii) 10 (second row) and (iii) 20 (third row). For different initial points and degrees of sharpness, Adam+CM converges to flatter minima more often than Adam+CG.

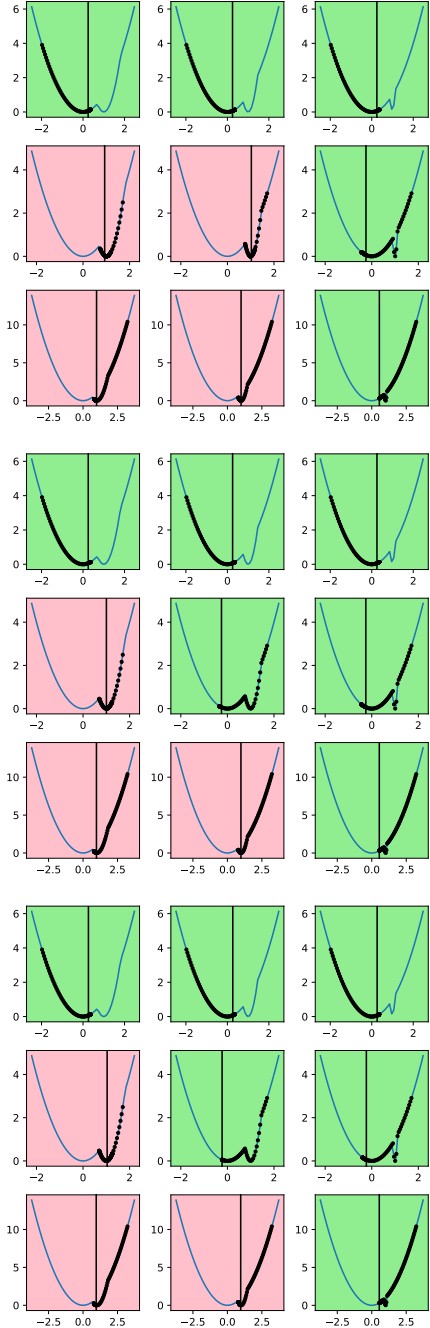
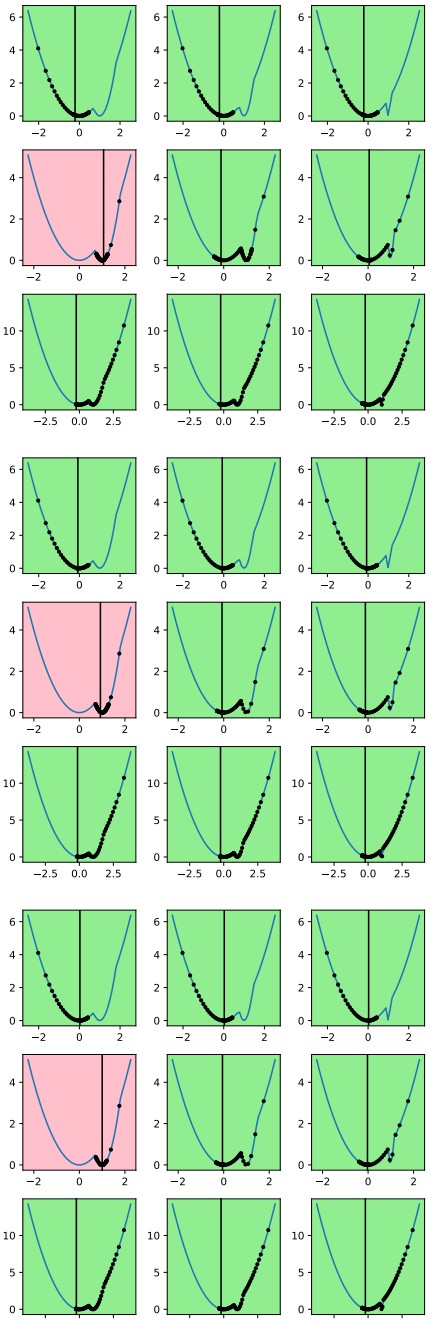

Figure 18: Following Figure 5, we compare trajectories of Adam+CG (left columns) and Adam+CM (right column) on Equation 9 with $d = 1$ where $\lambda$ is set to 0.99 and $C$: (i) 5 (first row), (ii) 10 (second row) and (iii) 20 (third row). For different initial points and degrees of sharpness, Adam+CM converges to flatter minima more often than Adam+CG.

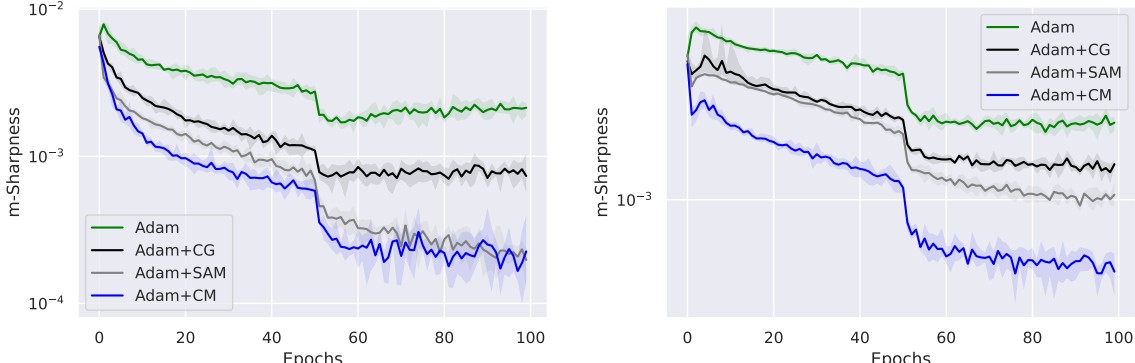

Figure 19: $m$-sharpness computed upon training ResNet34 on CIFAR10 (left) and CIFAR100 (right) with different optimizers on a fixed hyper-parameter setup. In both cases, Adam+CM has lower m-sharpness than other baselines.

| Learning rate | Validation Accuracy |
|:---:|:---:|
| 0.01 | $93.5_{\pm 0.3}$ |
| 0.001 | $\mathbf{93.7}_{\pm 0.2}$ |
| 0.0001 | $93.4_{\pm 0.2}$ |

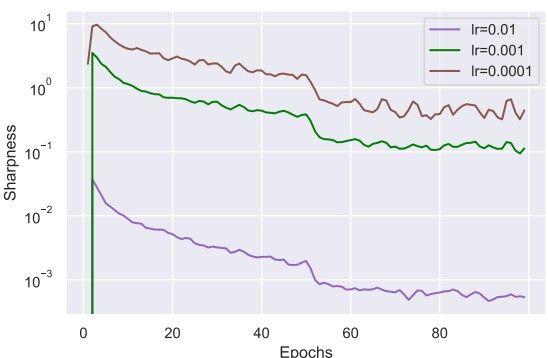

Table 14: Validation Accuracy of Adam with different learning rates on CIFAR10. The best result was obtained using a learning rate of 0.001.

Figure 20: Sharpness obtained using Adam with different learning rates on CIFAR10. These results indicate that a higher learning rate results in lower sharpness.

