# OpenReview forum: "Promoting Exploration in Memory-Augmented Adam using Critical Momenta"
_TMLR — Accepted by TMLR_

### Review · Reviewer_EuAX · 2024-03-29

**Summary Of Contributions:**

Thes paper provides an improvement over adam+cg and proposes adam+cm (critical momenta). The main innovation is that adam+cg, which is an exploratory step built on top of adam to escape sharp minimas, can have the drawback that when it encounters a large gradient in a sharp minima, it cancels with existent gradient buffers and provides no exploration. To improve over this, the authors apply the buffering on the momenta term instead of the gradient. Through analyzing trajectories on non-convex toy examples, they show that adam+cm can escape sharp minimas whereas other variants of adam + exploration strategies fail to do so. Furthermore, this method has a faster convergence rate on quadratic losses.

**Audience:**

Yes

**Claims And Evidence:**

Yes

**Requested Changes:**

I would like the authors to answer the 3 weakness points raised. Further answering the questions 1 and 2 are critical.

**Strengths And Weaknesses:**

Strengths:

1) The paper presents an interesting algorithm of improving generalization by escaping sharp minimas. Adam is well known to generalize worse than SGD, and improving it's generalization is critical.
2) The experiments on toy examples and large datasets show the applicability of this method well.

Weakness / Questions:

1) "Gradient cancellation primarily
occurs when existing buffer gradients are quickly replaced by high-magnitude gradients when the parameters
are near a sharp basin. " this argument does not seem to be generic. There is no guarantee that at sharp minima, the current gradient will oppose the existing buffer accumulation. It depends on the direction of divergence and may vary for different loss curves. I could counter-argue that the current gradient will be in the same direction as the existing buffer accumulation facilitating escape from sharp minimas. Curves in Figure-2 seem very specific to the Ackley loss surface.

2) I wonder methods like momentum and ADAM already take in the trajectory history into account. So, why would there be a need for additional exploration. Methods like SAM takes a flatter direction step from the current iterate and that is understandable. Infact there are works that explore extragradient exploration in top of momentum (https://openreview.net/forum?id=ZLVbQEu4Ab). Why shouldn't these methods be enough? Furthermore, just averaging weights in the trajectory are well known to improve generaliaztion (https://arxiv.org/abs/1803.05407). Having just model averging on top of Adam should work well.

3) Figure-4  shows signs that learning rate decay was used at around 250 epoch for Adam+CM, Adam+SAM+CM, Adam+CG, which is known to decrease the training loss abruptly. Infact, before 250 epoch, just Adam+SAM seem to have a lower loss. Can the authors confirm this? And if so, a fair comparison should be done without learning rate decay.

Questions:

1) Will increasing the learning rate in Adam help to escape sharp minimas. For plain Adam (without an explorations) with step size η and
β1 = 0.9, this stability threshold is 38/η  (https://arxiv.org/pdf/2207.14484.pdf).  Can lr be increased enough to match the sharpness levels in Figure-8?

2) How does the generalization of simple SGD compare with the generalization capabilities of exploration based Adam methods? I did not find any comparison with this.

---

> ### Author Response · Authors · 2024-04-25
>
> We thank the reviewer for their extensive evaluation. We especially appreciate their positive feedback on the algorithm and the analysis presented in our paper. Regarded the pointed weaknesses, please consider our responses below:
>
> **Reviewer’s Comment: “Gradient cancellation primarily occurs when existing buffer gradients are quickly replaced … I could counter-argue that the current gradient will be in the same direction as the existing buffer accumulation facilitating escape from sharp minimas. Curves in Figure-2 seem very specific to the Ackley loss surface”**
>
> **Reponse:**  Thank you for this insightful feedback. We agree that the overshooting phenomenon depends on the direction of divergence and varies for different loss curves. However, we argue that the gradient cancellation could still occur even when the sharpness is not across all dimensions as in Ackley. We verify this by plotting six consecutive gradient steps using Adam+CG and Adam+CM on the Goldstein–Price function. We have added the screenshots of these steps in Appendix A.1 (Fig 12). Similar to our previous observations on Ackley function, we find that the buffer quantities in CG are dominated by directions of higher sharpness (top-left and bottom-right). On the other hand, in CM, the buffer directions do not drastically change. As a result, although CG initially escapes the sharp basin, it eventually gets stuck in the sharp basin whereas CM finds the surface that contains the global minima.
>
> **Reviewer’s Comment: “I wonder methods like momentum and ADAM already take in the trajectory history into account. So, why would there be a need for additional exploration. Infact there are works that explore extragradient exploration in top of momentum. Why shouldn't these methods be enough?”**
>
> **Reponse:**  While momentum and Adam take the trajectory history into account, we argue that they converge to suboptimal minima near the initialization point. In particular, using toy examples, we show that Adam tends to get stuck in nearby minima in Fig 2, Fig 4, Fig 5 (left), and Fig 6, etc. In the case of supervised learning, we also show the same phenomenon in terms of distance traveled by the parameters using Adam (Fig 8 top-right). Furthermore, the minima found using Adam are also shown to be sub-optimal (Table 2) and sharp (Fig 8 top-left). Hence, we propose to use critical momenta to efficiently explore the loss surface and avoid getting stuck at sharper minima, finding a more generalizable solution. In addition, following the reviewer's suggestion, we provide the adaptation trajectories of the referred existing method MEG (momentum extragradient) on different loss surfaces in the revised manuscripts in Appendix A.2.1 (Fig 15). We compare the trajectories of Adam+MEG with other variants and show that CM still exhibits better exploration. When evaluated on CIFAR10 dataset, Adam+MEG achieves the accuracy of $93.8 (\pm 0.3) %$ which is lower than both Adam+CM ($94.0 (\pm 0.3) %$) and Adam+SAM+CM ($94.4 (\pm 0.4) %$)
>
> On the other hand, stochastic weight averaging (SWA) is proposed to specifically improve generalization of the model rather than adding exploration capabilities. Therefore, methods like SWA remain complementary to our proposed method.
>
> **Reviewer’s Comment: “Figure-4 shows signs that learning rate decay was used ... known to decrease the training loss abruptly ... a fair comparison should be done without learning rate decay.”**
>
> **Reponse:**  Thank you for your comment. Before the 250 epoch, Adam+SAM had a lower loss because it found a nearby minimum and got stuck as shown in Fig 4 and Fig 12 (orange color trajectories). On the other hand, both Adam+CM and Adam+SAM+CM escape that minima and keep on exploring the noisy surface as a result of which has a higher loss at times. However, due to extensive exploration, it eventually arrived near the basin containing the global minima. Therefore, reducing the learning rate helps in settling down to the global minimum and stopping further exploration.
>
> Following the reviewer's suggestion, we ran a similar experiment with the default hyper-parameters and without learning rate decay. We have added the corresponding plot showing the training loss comparison in Appendix A.2.1. In this case, we observed that Adam+CM oscillates near the global minima because the surface around the global minima is noisy as well.

---

> ### Author Response · Authors · 2024-04-25
>
> **Reviewer’s Comment: “Will increasing the learning rate in Adam help to escape sharp minimas. ... Can lr be increased enough to match the sharpness levels in Figure-8?”**
>
> **Reponse:**  We have indeed observed that increasing the learning rate will help in reducing the sharpness but this results in worse generalization performance in deep learning experiments. Therefore we show the best performing setup in Fig 8.
>
> In response to the reviewer's question, we have added a figure in Appendix A.3.2 showing the empirical sharpness for different learning rates in Adam. We also provide the generalization performances of the corresponding setups and show that Adam learning rate of 0.001 performs better.
>
> **Reviewer’s Comment: “How does the generalization of simple SGD compare with the generalization capabilities of exploration based Adam methods?.”**
>
> **Reponse:**  We have reported the generalization performance of SGD in Table 3. We find that after an extensive hyper-parameter search Adam and its variants perform better than both SGD and SGD+CM.
>
> We hope that the above responses address the reviewer’s concerns. We revised our paper with the requested changes highlighted in blue. If there are more comments or questions, please let us know.

---

### Review · Reviewer_kzdk · 2024-04-14

**Summary Of Contributions:**

The paper proposes a new algorithm within the memory-augmented framework applied to Adam called Adam with Critical Momenta (Adam + CM). The effect of CM is shown in several toy examples showing that Adam with CM is able to find stationary points with flatter surface than existing variants of Adam. Numerical experiments are provided to illustrate the advantage of CM in different benchmarks including language modeling, image classification, and online learning.

**Audience:**

Yes

**Claims And Evidence:**

Yes

**Requested Changes:**

- I think the equation references in Algorithm 1 can be improved. E.g.: Update 1nd moment should point to equation 4. Update parameter $\theta_t$ should point to equation 7?
- I think it is worth to explicitly define $h$ (probably eigenvalues of $H$) in section 3.2
- I suggest having the same order and color for 2 plots in Figure 4.
- For completeness, I would explicitly describe how CG and CM are integrated into SGD. Even though it can be infered from how CG is applied to Adam.
- There are few cases where combing CM with SAM does not perform as well, or maybe worse than their individual technique. I wonder if there is any intuition on that.
- In several experiments the variant is a bit high which includes the interval of other variant (despite higher mean) so we can't really conclude Adam+CM is indeed better. E.g: CIFAR10, CIFAR100 in Table 2, 5-dataset in Table 4.

**Strengths And Weaknesses:**

Strenghts:
- The idea of CM is simple yet effective by applying same techniques as CG into the first-order momentum term of Adam instead of the gradient.
- The paper points out the drawback of gradient cancellation when using Adam with CG and propose CM to address that.
- The authors do a good job of illustrating how Adam with CM can lead to better stationary points compared to existing variants by looking at several toy examples in Section 4.
- Numerical experiments are very extensive, there are many ablation studies on effect of key parameters such as buffer size vs local sharpness.

Weakness:
- The convergence analysis only considers the quadratic losses which is somewhat simple. I hope to see analyses for broader classes of function.

---

> ### Author Response · Authors · 2024-04-25
>
> We’d like to thank the reviewer for the evaluation and acknowledging the extensive analysis provided in our paper. We also appreciate the comments and provide our responses below:
>
>
> **Reviewer’s Comment: “The convergence analysis only considers the quadratic losses which is somewhat simple. I hope to see analyses for broader classes of function.”**
>
> **Reponse:**  The quadratic assumption can be (slightly) relaxed in our analysis by assuming the loss Hessian matrices at computed along the optimization trajectory all commute with each other (for example, by considering a loss with diagonal Hessian). Considering more general function classes would indeed be very interesting,  but we believe this is a much more difficult problem. We would also like to re-iterate that the main goal of our paper is to enable efficient exploration in adaptive optimizers and we propose a method that not only explores but also finds flatter and generalizable solutions.
>
> **Reviewer’s Comment: “I think the equation references in Algorithm 1 can be improved.”**
>
> **Reponse:**  Thank you for pointing this out. We have updated this part in the revised manuscript and highlighted it in blue.
>
> **Reviewer’s Comment: “I think it is worth to explicitly define h (probably eigenvalues of H) in section 3.2”**
>
> **Reponse:**  We have incorporated the above suggestions in section 3.2 in the revised manuscript and highlighted it in blue. In particular, we added the following:
>
> >  Let $h$ denote the eigenvalues of the Hessian matrix $H$. Note that the singular values of the block matrix $A$ are the same as those of the matrices $A_h$, obtained by collapsing the blocks of $A$ as follows: replace $I$ by $1$ and $H$ by $h$.
>
> **Reviewer’s Comment: “I suggest having the same order and color for 2 plots in Figure 4”**
>
> **Reponse:** We appreciate the suggestion made by the reviewer about maintaining the order and color in the plots. While we follow the same order and color for all the line plots presented in the manuscript (Fig. 4 Left and Fig. 8), we used a different color and linestyle for Adam+CG and Adam+SAM for the trajectory plots on toy loss surfaces for better contrast against the contour plots (Fig. 4 Right, Fig. 13 etc.).
>
>
> **Reviewer’s Comment: “For completeness, I would explicitly describe how CG and CM are integrated into SGD. Even though it can be infered from how CG is applied to Adam”**
>
> **Reponse:**  We have added the following details about SGD variants in section 5.3:
>
> > To implement SGD+CG, we follow McRae et al. (2022) and replace $g_t$ with aggregation of $g_t$ and the buffer gradients $g_c$:
> >   $\theta_{t+1}$ = $\theta_t$ - $\eta$ $[g_t + (1/C) \sum g_c]$
> > Similarly, for SGD+CM, we aggregate the momentum using equation 4 and update the parameters using equation 8.
>
> **Reviewer’s Comment: “There are few cases where combinig CM with SAM does not perform as well, or maybe worse than their individual technique.”**
>
> **Reponse:**  This is a good point. We speculate that this happens in contexts where SAM limits the amount of exploration brought by CM (e.g., by favoring convergence in flatter but higher loss regions).  Furthermore, it is worth noting that we also observed that, for models trained on  CIFAR100 dataset, CM with slightly higher sharpness performed better than SAM+CM (Ref. Table 2 and Fig 8). This suggests that, in this particular case, a sharper solution would generalize better.
>
>
> **Reviewer’s Comment: “the variant is a bit high which includes the interval of other variant (despite higher mean) so we can't really conclude Adam+CM is indeed better”**
>
> **Reponse:** Indeed, this is true but we would like to argue that the overlap mainly occurs on smaller datasets such as CIFAR10. On the other hand, in the setups based on more complex dataset such as ImageNet, additional exploration helps in navigating the loss landscape as Adam+CM has a clear gain in performance.
>
>
> We hope that the above responses address the reviewer’s concerns. We revised our paper with the requested changes highlighted in blue. If there are more comments or questions, please let us know.

---

### Review · Reviewer_q45D · 2024-04-15

**Summary Of Contributions:**

This paper proposed a variant of memory-augmented optimizer (McRae et al. 2022), with replacement of gradient by the momentum and applied specifically in Adam-class algorithms. The main idea is to use a memory buffer to store the past history of momentums, aka `critical momenta`, which alleviate the issues of `gradient cancellation` induced by `critical gradient` proposed in McRae et al. (2022). This allows the algorithm to escape sharp local curvature and seek for flatter neighborhood of minima.

**Audience:**

Yes

**Broader Impact Concerns:**

I do not see a `broader impact statement` being enclosed in the current version of the paper. However, as I see that the paper mainly concerns a very specific improvement in enhancing the performance of a very well known optimizer, i.e., Adam, and thus I have no concerns of any negative impact.

**Claims And Evidence:**

Yes

**Requested Changes:**

$\textit{Note}$: this section covers both questions and requested changes.

$\textbf{Q.1}$. In McRae et al. (2022)., author explicitly explains that using memory buffer with past gradient captures the curvature. Is there anyway besides the specific/simple example showing `section 3.2` to generalize the idea? or if author had different ideas of how the CM helps rather than better capturing the curvature, would it be possible to explain it further with generalized case?

$\textbf{Q.2}$. Compared to McRae et al. (2022)., Can I assume the equivalent cost of computation and space complexity? Is it possible to add complexity analysis to showcase the merit of the proposed CM method?

$\textbf{Q.3}$. In `section 5.3`, authors briefly mention applying CM with SGD. Would it be possible (or appropriate) to address the different memory-augmented SGD variants in more details? It seems that the current version of paper has no mention of these elsewhere.

$\textbf{Q.4}$. Given the emphasis of 'sharpness' of this paper, would it be better to give more explanation and/or define the sharpness more formally, so to benefit the general audience? Also, for 'm-sharpness' seen in Appendix, it is clear that the definition/idea was from pre-existing literature, and could be very beneficial to add more context in the current version of the paper.

**Strengths And Weaknesses:**

$\textbf{Strength}$

$\textbf{S.1}$. The proposed critical momenta is a direct enhancement on the critical gradient version of the memory-augmented Adam. The motivation is strong with very intuitive explanation of the issues, i.e., gradient cancellation, caused by critical gradient. The approach of resolving the issue and hence improve the performance is very well presented and novel.

$\textbf{S.2}$. There are a considerable amount of very intuitive and insightful explanation on how to use memory buffer to better leverage the curvature, i.e., second-order information, what kind of issues existed in pre-existing literature of memory-augmented optimizers, and how to address the issues. It does an excellent job on educating the audience and presenting the novel idea of using critical momenta.

$\textbf{S.3}$. The brief technical analysis is very insightful and convincing in demonstrating the proposed approach given in convex context. Meanwhile, there are a lot of insightful and intuitive empirical analysis to further explain and present the idea.

$\textbf{S.4}$. The main empirical study is very extensive. While the technical insight mainly focuses on convex case, the main experiment focuses mainly on up-to-date relevant nonconvex problems in supervised learning, such as token classification in language modeling and image classification, as well as online learning setting.



$\textbf{Weakness}$:

$\textbf{W.1}$. Compared to the empirical analysis, the technical analysis, especially regarding the convergence analysis, is rather weak. There lacks a formal convergence analysis in the proposed strongly convex case, and whether a convergence can be achieved or not in a (specific or general) nonconvex setting is unknown.

$\textbf{W.2}$. While the technical analysis reveals how the curvature is better captured through CM and how the 'flatter sharpness' is realized. It is lacking a formal analysis, on defining i) the relevant metrics, such as 'how flat is flatter?' and 'what does flatness mean' ii) how the curvature is better captured or captured in a certain way, resulting an improved behavior over CG version.

---

> ### Author Response · Authors · 2024-04-25
>
> We’d like to thank the reviewer for the positive evaluation of our paper about the intuition, explanation and empirical analysis presented in our method. We also appreciate the suggestions  and provide our responses below:
>
> **Reviewer’s Comment: “Compared to the empirical analysis, the technical analysis, especially regarding the convergence analysis, is rather weak. There lacks a formal convergence analysis in the proposed strongly convex case, and whether a convergence can be achieved or not in a (specific or general) nonconvex setting is unknown.”**
>
> **Reponse:**  We would like to clarify that Section 3.2  does indeed provide a formal convergence analysis in the quadratic setting. Recasting CM as a multistep linear system, we leverage the classical worst-case bound of linear systems, which guarantees convergence whenever the spectral radius $\rho(\alpha, \beta)$ is strictly lower than one. In this context, the problem is to show that the parameters $\alpha, \beta$ can be tuned to ensure convergence, i.e. $\rho*$ : = $min_{\alpha, \beta}~~\rho(\alpha, \beta) < 1$. We show numerically that this is the case  and compare the optimal rate of convergence with classical momentum in Fig 3.
>
> We agree that analogous results for the nonconvex case would be extremely interesting, yet we believe this is a much more difficult problem. We would also like to highlight that the primary contribution of our paper is to approach the optimization problem from the perspective of the loss surface exploration issue and we draw inspiration from memory-augmented optimizers to address this.
>
>
>
> **Reviewer’s Comment: “It is lacking a formal analysis, on defining i) the relevant metrics, such as 'how flat is flatter?' and 'what does flatness mean' ii) how the curvature is better captured or captured in a certain way,... Is there anyway besides the specific/simple example showing section 3.2 to generalize the idea? … would it be better to give more explanation and/or define the sharpness more formally?”**
>
> **Reponse:**  The goal of our work is not to explicitly capture curvature, but to promote efficient exploration of the loss surface.  Specifically, critical momenta are designed to mitigate the gradient cancellation problem of previous memory-based optimizer CG (McRae et al. (2022)). Our empirical analysis shows that, as a result, CM finds flatter solutions (as measured by standard curvature proxies such as the largest Hessian eigenvalue) – which in turn tends to improve overall performance. We are happy to continue the discussion if our response does not adequately address the reviewer’s comments.
>
>
> In response to the reviewer's feedback,  we made the definitions of sharpness/flatness clearer in our paper. In particular, we have added the following in section 3.2:
>
> > Following the literature (Kaur et al., 2023), we use maximum eigenvalue ($h_{max}$) of the Hessian $H$ as the indicator of sharpness in the rest of the paper. As $h_{max}$ increases, the surface becomes sharper.
>
> We have also added more details about m-sharpness in section A.3.1:
>
> > The $m$-sharpness is defined as:
> >  $(1/n)$ $\sum_{M \in D}$ $max_{||\epsilon|| \leq r}$ $(1/m)$ $\sum_{s \in M}$ $L_s(\theta + \epsilon) - L_s(\theta)$,
> > where $D$ represents the training dataset, which is composed of $n$ mini-batches $M$ of size $m$ and $r$ is a hyper-parameter set to $0.05$ by default.
>
>
> **Reviewer’s Comment: “Can I assume the equivalent cost of computation and space complexity?”**
>
> **Reponse:**  Yes, CM uses the equivalent cost of computation and space complexity as CG (McRae et al. (2022)). However, after performing a hyperparameter grid search over the buffer size, we observed that CM consistently requires a smaller buffer size than CG and obtains better performance (Table 9 and Table 10). We have specified it in the Appendix A.2.2 as well in the revised manuscript.
>
>
> **Reviewer’s Comment: “Would it be possible (or appropriate) to address the different memory-augmented SGD variants in more details?”**
>
> **Reponse:**  Thank you for pointing this out. We have added the following details about SGD variants in section 5.3:
>
> > To implement SGD+CG, we follow McRae et al. (2022) and replace $g_t$ with aggregation of $g_t$ and the buffer gradients $g_c$:
> >   $\theta_{t+1}$ = $\theta_t$ - $\eta$ $[g_t + (1/C) \sum g_c]$
> > Similarly, for SGD+CM, we aggregate the momentum using equation 4 and update the parameters using equation 8.
>
>
> We hope that the above responses address the reviewer’s concerns. We have also revised our paper with the requested changes highlighted in blue. If there are more comments or questions, please let us know.

---

### Decision · Action_Editor_1K8f · 2024-05-29

**Recommendation:** Accept with minor revision

**Comment:**

### Comment on the recommendation
The reviewers agree that the work is well-motivated, the paper is clearly written, and the claims are supported by high-quality empirical results. (Please also see **Claims And Evidence**.)

The authors addressed the major concerns raised by the reviewers with the replies and the revised manuscript.

The contributions are highly relevant to the TMLR community, presented in a good quality. Hence, I recommend acceptance.

### Requests to the author for the final version
* Please perform minor revisions based on the revised version uploaded on Apr 25.
* I suggest that the authors briefly mention the limitation of the theory part, e.g., in Section 6 as this was the biggest concern of the reviewers.
* Please consider improving Figure 2. The contours are nearly invisible to me.
* I think Figure 1 should be for Table 1 in the sentence "Results in Figure 1 show [...]" on p.7.
* Please add punctuation marks after equations.
* Typo: worse-case --> worst-case

**Audience:**

As the reviewers expressed in their comments, the paper's contributions do interest the TMLR's audience. Indeed, the proposed optimization trick is quite generally applicable to many machine learning methods.

**Claims And Evidence:**

The paper proposes an enhancement of an optimization trick called the _Critical Gradients (CG)_ (McRae et al., 2022). The proposed method aggregates the history of momenta instead of that of gradients, hence called the _Critical Momenta (CM)_. The authors claim that gradients around a sharp local minimum have diverse directions and cancel out each other when aggregated with the CG, ending up being stuck. The proposed CM, on the other hand, avoids such gradient cancellation by using momenta, helping the optimizer escape sharp local minima.

The reviewers agree that these claims are well-supported by extensive numerical experiments from simple toy examples to more complex real-world applications. The results indeed show that the CM often leads an optimizer to escape sharp local minima and converge to flatter one.

The paper also provides a theoretical convergence result for a simple quadratic function and simpler optimization updates.  Although it is a relevant result, the reviewers mentioned that this contribution is limited in that the studied scenario is much more simplified compared to the actual use shown in experiments.